# ParallelSpec: Parallel Drafter for Efficient Speculative Decoding

## Abstract

Speculative decoding has proven to be an efficient solution to large language model (LLM) inference, where the small drafter predicts future tokens at a low cost, and the target model is leveraged to verify them in parallel. However, most existing works still draft tokens auto-regressively to maintain sequential dependency in language modeling, which we consider a huge computational burden in speculative decoding. We present ParallelSpec, an alternative to auto-regressive drafting strategies in state-of-the-art speculative decoding approaches. In contrast to auto-regressive drafting in the speculative stage, we train a parallel drafter to serve as an efficient speculative model. ParallelSpec learns to efficiently predict multiple future tokens in parallel using a single model, and it can be integrated into any speculative decoding framework that requires aligning the output distributions of the drafter and the target model with minimal training cost. Experimental results show that ParallelSpec accelerates baseline methods in latency up to 62% on text generation benchmarks on Medusa and 9-17% on EAGLE. It also achieves 2.84× overall speedup on the Llama-2-13B model using third-party evaluation criteria.

## 1 Introduction

Large language models (LLMs) such as GPT-4 (OpenAI, 2023) and Llama (Touvron et al., 2023) have shown dominant abilities across various domains, such as question answering (Zhuang et al., 2023), code synthesis (Rozière et al., 2023), machine translation (Zhang et al., 2023a) and beyond. However, their auto-regressive nature requires multiple forward passes on models with billions or trillions of parameters, bringing substantial inference latency, thus prohibiting real-time applications. In the pursuit of accelerating LLM inference, various strategies have been explored, including utilizing model sparsity (Liu et al., 2023; Sun et al., 2024; Schuster et al., 2022; Cai et al., 2024a), exploiting redundancy in KV Cache (Cai. et al., 2024; Zhang et al., 2023b; Li et al., 2024a), and distilling model capabilities to smaller models (Gu et al., 2024; Agarwal et al., 2024). While these approaches can lead to faster inference, they often come at the cost of reduced generation quality and do not preserve the generation distributions of the original models.

Speculative decoding (SD) (Leviathan et al., 2023; Chen et al., 2023a) has been proposed as one of the compelling alternatives to auto-regressive generation in a lossless manner. The key motivation behind SD is to utilize a low-cost small model to generate draft tokens efficiently and then use the target model to verify them in parallel to ensure sampling integrity, known as *draft-then-verify* framework. While promising, we observe that draft models in most *draft-then-verify* frameworks still generate token by token, resulting in a low arithmetic intensity during the drafting stage. Moreover, the forward latency of the drafting stage still grows linearly with respect to the draft length, *i.e.*, the number of tokens each draft step generates. As empirically profiled in the right part of Figure 1, the draft latency still accounts for a significant proportion of the overall SD latency.

Researchers have spotted the draft latency issue and proposed several solutions to alleviate it by dynamically determining the draft length either through learnable policy (Huang et al., 2024a), confidence-guided heuristics (Li et al., 2024c), or optimized draft tree structures (Wang et al., 2024a). Nevertheless, these solutions do not tackle the underlying issue of drafting latency scaling linearly with the draft length, but operate only for maximally reducing the compute wasted in drafting tokens that are unlikely to be accepted.

Figure 1: Illustration of PARALLELSPEC. The Left part of the figure has been revised to highlight the difference between the two drafting styles. **Left:** comparison between auto-regressive drafting and our proposed parallel drafting. Blocks in green indicate normal draft tokens. Blocks in yellow denote the mask tokens used to prompt the draft model to generate multiple future tokens in a single forward pass. **Right:** wall time trace diagrams for two drafting styles integrated with EAGLE (Li et al., 2024b) in two rounds of speculative sampling, given the assumption that both drafting styles have the same speculation accuracy on the prefix sequence.

To fundamentally solve the draft latency issue, we propose building a parallel-decoding drafter as the replacement for auto-regressive drafters in popular SD frameworks. Unlike Medusa-style frameworks (Cai et al., 2024b; Ankner et al., 2024) that rely on separate language model heads to decode future tokens, we propose to use a single lightweight model to decode the next $k$ tokens simultaneously. We argue that using a single model for multi-token prediction can effectively leverage parameter sharing to achieve efficient drafter alignment rather than learning several independent language model heads. The latter design would struggle even more with memory and computation in the large vocabulary size (128,000+) of recently introduced language models such as Llama-3 (AI@Meta, 2024). For efficient multi-token alignment training, we introduce a group-wise parallel training strategy that mitigates possible training-inference mismatches by dynamically adjusting the attention mask, positional indexes, and token layout.

Our method still adheres to the *draft-and-verify* framework at inference time: At each draft step, the drafter generates $k$ tokens with a single forward pass, and then they are sent to the target model for parallel verification. Since our method incorporates token-level parallelization in the drafting stage, both stages in the SD pipeline now benefit from this parallelization. Therefore, we name our approach PARALLELSPEC. PARALLELSPEC works as an individual module, ready to replace any drafter in existing SD frameworks that require distilling drafters from their target models. We experiment with the popular speculative decoding framework Medusa (Cai et al., 2024b) and state-of-the-art solution EAGLE (Li et al., 2024b) by replacing their drafters. Experimental results show that PARALLELSPEC is able to bring consistent acceleration improvement in all task domains and different combinations of models. For instance, incorporating PARALLELSPEC into Vicuna-7B Medusa increases the average speedup ratio from $1.42\times$ to $2.31\times$, leading to a 62.7% relative improvement. This empirically validates the superiority of parallel drafter design. PARALLELSPEC integration with EAGLE also achieves extra speedups across all target model settings, ranging from $2.55\times$ to $2.84\times$, with a relative improvement ranging from 9% to 17%. In summary, our key contributions are as follows:

- We propose PARALLELSPEC as a parallel multi-token drafter to replace the auto-regressive drafter design in existing SD frameworks that require aligning drafters with their target models.

- We design a group-wise training strategy that allows efficient and accurate parallel drafter training.

- We integrate the proposed method into two popular SD frameworks. Extensive experiments demonstrate the compatibility and performance superiority of PARALLELSPEC.

## 2 RELATED WORKS

**Accelerating Large Language Model (LLM) Inference** has attracted considerable research attention from both machine learning system and natural language processing communities and even led

the trend of hardware-software co-design. These research efforts include model compression (Sun et al., 2024; Huang et al., 2024b; Ma et al., 2023), novel architecture design (Gu & Dao, 2023; Peng et al., 2023) and hardware optimization (Dao et al., 2022; Hong et al., 2023). However, some of these methods could lead to generation discrepancies compared to the target model, representing a trade-off between model performance and inference speed. We consider those methods that cannot generate with the target model's original distribution as lossy acceleration methods.

**Speculative Decoding (SD)** (Leviathan et al., 2023; Chen et al., 2023a) arises as one of the lossless acceleration methods for LLM inference. It is based on the observation that the latency bottleneck of LLM inference is brought by memory bandwidth instead of arithmetic computation. SD alleviates the bandwidth bottleneck by utilizing a small model to draft multiple tokens and verifying them in parallel with the *target* model, thereby reducing the frequency of language model calls and decreasing the memory access density during decoding. The community has witnessed many improvements in efficiently and accurately drafting tokens. Self-speculative decoding methods (Hooper et al., 2023; Elhoushi et al., 2024; Zhang et al., 2024a; Bhendawade et al., 2024) do not explicitly rely on draft models but use some of the intermediate layers of the target model to draft. Medusa-style methods add independent (Cai et al., 2024b) or sequential (Ankner et al., 2024) decoding heads on the target model to draft tokens. Lookahead decoding and its variants (Fu et al., 2024; Zhao et al., 2024) use n-gram trajectory as drafts. DistillSpec (Zhou et al., 2024) leverages knowledge distillation to closely align distributions between the draft and target models. PEARL (Liu et al., 2024) proposes two-stage verification to alleviate the mutual waiting problem. Apart from efficient draft methods, token tree verification (Miao et al., 2024; Sun et al., 2023) has been widely adopted for verifying top candidate sequences that share common prefixes in parallel. Specialized SD frameworks for long-context generation (Chen et al., 2024a;b), retrieval-augmented generation (He et al., 2024; Wang et al., 2024b; Zhang et al., 2024b) and beyond (Chen et al., 2023b) have been proposed to better fit individual use cases.

**Parallel Decoding** was first known for its efficiency in machine translation system (Ghazvininejad et al., 2019) and code generation (Gloeckle et al., 2024) as an alternative to auto-regressive generation. However, its usage in SD frameworks remains under-explored. Cai et al. (2024b) and Stern et al. (2018) utilize parallel language model heads to predict multiple tokens at different positions. Santilli et al. (2023) proposed to use fixed-point iterations to replace auto-regressive decoding. Monea et al. (2023); Yi et al. (2024) pioneered the use of parallel decoding in SD, but it is limited to a self-speculative framework and results in different generation sampling. BiTA (Lin et al., 2024) proposed using prompt tuning to train a small number of prompt parameters on a frozen target LM for semi-autoregressive generation. Wu et al. (2024) suggested using trainable linear projection to regress intermediate hidden states of target models, thereby enabling multi-token prediction. However, these methods either fail to effectively learn the draft distribution due to the limited number of learnable parameters or cannot achieve lossless acceleration due to the method design.

## 3 BACKGROUND: SPECULATIVE DECODING

**Notation.** Speculative decoding (SD) frameworks maintain two models: the *target* model, denoted as $\mathcal{M}_{\mathrm{T}}$, is the one which we want the SD frameworks to sample from; the *draft* model, denoted as $\mathcal{M}_{\mathrm{D}}$, is the one that proposes candidate tokens which are later being verified by the *target* model. Let $x_{<t}$ be the prompt sequence we are running the SD framework on, $p(x_t \mid x_{<t})$, $q(x_t \mid x_{<t})$ be the inference distribution of $\mathcal{M}_{\mathrm{T}}$ and $\mathcal{M}_{\mathrm{D}}$ given the prompt $x_{<t}$, respectively. We use the denotations of $p(y_t)$ and $q(y_t)$ to indicate $p(y_t \mid x, y_{<t})$ and $q(y_t \mid x, y_{<t})$ whenever they do not lead to confusion.

**Speculative Decoding Procedures.** One round of speculative decoding can be divided into the drafting and verification stages, each governed by the corresponding model. The drafting stage auto-regressively calls $\mathcal{M}_{\mathrm{D}}$ to sample $\gamma$ candidate token distributions $q(y_t), \ldots, q(y_{t+\gamma-1})$. The verification stage calls $\mathcal{M}_{\mathrm{T}}$ once to sample $\gamma$ distributions from the target model, $p(y_t), \ldots, p(y_{t+\gamma-1})$, given $y_{t+i}$ is the concatenation of $y_{<t}$ and drafted token sequence $x_1, \ldots, x_i$, where $x_k \sim q(y_{t+k})$. The verification stage determines whether the token $y_{t+i}$ is accepted via speculative sampling, where its acceptance rate $\alpha_i$ is defined as:

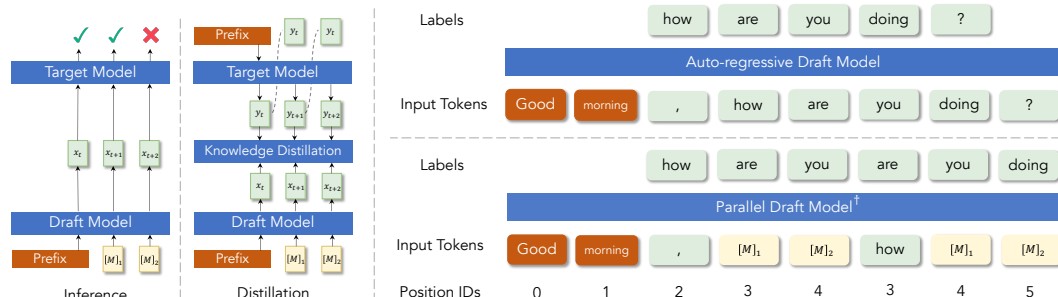

Figure 2: Illustration of parallel drafter inference, training, and the difference between training auto-regressive drafter and parallel one. **Left:** Parallel drafter proposes multiple candidate tokens with a single forward pass. **Middle:** Training the parallel drafter to align with the target model is a process of knowledge distillation (KD). **Right:** The input, labels, and position indices for training a parallel drafter need special treatment. † refers to Figure 3 for the special attention mask design of parallel training.

$$\alpha_i = \begin{cases} 1 & p\left(y_{t+i}\right) \geq q\left(y_{t+i}\right) \\ \frac{p(y_{t+i})}{q(y_{t+i})} & p\left(y_{t+i}\right) < q\left(y_{t+i}\right) \end{cases} \tag{1}$$

If the token $y_{t+i}$ is rejected before $\gamma$ candidate tokens are all accepted, the remaining draft tokens will be discarded, and $y_{t+i}$ will be resampled from $\max\left(0, p\left(y_{t+i}\right) - q\left(y_{t+i}\right)\right)$. Otherwise, drafted tokens are all accepted and SD samples an extra token from $y_{t+\gamma}$ and appends it to the end of the sequence. Each round of speculative decoding generates at least 1 and at most $\gamma + 1$ tokens, and Leviathan et al. (2023) theoretically proves that the sequence from SD and the sequence from the target model follow the same distribution.

**Average Acceptance Length.** One important efficiency metric to measure an SD system is the acceptance rate of drafted tokens in each round of speculative decoding. Since each drafting step takes a constant amount of time and each round of conventional SD takes the same drafting steps, the overall efficiency of an SD system will be determined by the average acceptance length $\tau$ measured on some prompt sequences.

**Token Tree Verification.** Prior studies (Miao et al., 2024; Leviathan et al., 2023) suggest that verifying multiple candidate sequences within the same verification step could greatly improve the expected acceptance length. This is achieved by the tree attention mechanism. By properly arranging the top predicted tokens at different token positions and manipulating the attention mask based on a tree structure, we enable the processing of multiple candidate sequences with only one verification step. We refer to the details of token tree verification in Appendix A.3.

## 4 METHODOLOGY

In this section, we describe our parallel drafting method (§4.1), the algorithm that preserves the output distribution of the target model with parallel drafter (§4.2) and its integration into popular speculative sampling methods (§4.3).

### 4.1 PARALLEL DRAFTING

**Inference.** Let $\mathcal{M}_D^\theta$ be the parallel drafter parameterized by $\theta$, and $q^\theta\left(x_t, x_{t+1}, \ldots, x_{t+k} \mid x_{<t}\right)$ be the multi-token output distribution of the parallel drafter. Naive draft models do not support predicting multiple tokens in a single forward pass. In order to equip a drafter with such abilities, we propose to use customized [MASK] tokens as prompt tokens to produce contextualized token-level representations that are used to enforce multi-token training. The parallel drafter introduces $k$ special tokens in its vocabulary, [MASK]$_1, \ldots,$ [MASK]$_k$. At each drafting step where the drafter is invoked, $k$ special tokens are concatenated after the original input sequence $x_{<t}$. Apart from

the last token representation that is used to decode the next token $x_t$, the last-layer representations corresponding to [MASK] tokens are leveraged to sample future tokens $x_{t+1}, \ldots, x_{t+k}$. The left part of Figure 2 demonstrates how the parallel draft model proposes 3 future tokens with 2 [MASK] prompt tokens.

Figure 3: Attention mask illustration of parallel drafter training. ✓ denotes activated attention. ✗ denotes attention suppressed to prevent access across parallel groups. -100 denotes ignored tokens in the target sequence that do not contribute to training loss. Blocks with yellow and the legend illustrate one of the next-next token prediction training objectives.

**Group-wise Parallel Training.** Given the access to either the ground truth future tokens $y_t, y_{t+1}, \ldots, y_{t+k}$ or their output distributions of the *target* model $p(y_t), p(y_{t+1}), \ldots, p(y_{t+k})$, training a parallel drafter is a process of knowledge distillation (KD) in either online or offline setting (Zhou et al., 2024), which we defer to §4.3 for detailed discussion. This process is not as trivial as training an auto-regressive drafter, where teacher-forcing supervision is enforced via *shift-one-token* in labels as depicted in the upper right of Figure 2. Simply shifting the label position by $k$ tokens would result in discrepancies between training and inference. Therefore, We carefully design a group-wise training paradigm that eliminates training-inference mismatch by manipulating the token layout, attention masks, and position indices, illustrated in the lower right of Figure 2. Specifically, we introduce the concept of *parallel group*, illustrated in Figure 3, where each *parallel group* in the source sequence consists of an input token and several [MASK] tokens. At each training step, a customized causal attention mask guarantees that all training token pairs ignore [MASK] tokens from previous *parallel group*s to ensure the model behaves the same in training and evaluation.

## 4.2 SPECULATIVE SAMPLING WITH PARALLEL DRAFTING

In order to preserve the output distribution of the target model as standard speculative sampling does, following Monea et al. (2023), we present a modified version of speculative sampling (Chen et al., 2023a) that drafts with a parallel decoder and verifies with the target model using token tree verification method (Miao et al., 2024), detailed in Algorithm 1.

## 4.3 INTEGRATION WITH POPULAR SPECULATIVE DECODING FRAMEWORKS

PARALLELSPEC can be inserted into any speculative decoding (SD) framework that requires aligning output distributions of the drafter and the target model. We choose two popular SD frameworks as testbeds, Medusa (Cai et al., 2024b) and EAGLE (Li et al., 2024b).

**Medusa** leverages an *offline* knowledge distillation method, where cross-entropy loss between Medusa heads and ground truth tokens is used for alignment. Specifically, $K$ extra decoding heads are added to decode the last hidden states of the target model, and the $k$-th head is used to predict the $(t + k + 1)$-th token given a prefix sequence of length $t$. The final training objective of Medusa is expressed as:

$$\mathcal{L}_{\text{MEDUSA}} = \sum_{k=1}^{K} -\lambda_k \log p_t^{(k)} \left( y_{t+k+1} \right), \tag{2}$$

where $\lambda_k$ is a coefficient to balance learning difficulties, $p_t^{(k)}$ is the output distribution of the $k$-th head and $y_{t+k+1}$ is the oracle token.

---

**Algorithm 1** Speculative Sampling with Parallel Draft Models and Token Tree Verification

---

Given $k$ special tokens $[\texttt{MASK}]_1, \ldots, [\texttt{MASK}]_k$ and minimum target sequence length $T$.
Given the target model distribution $p(\cdot|\cdot)$, the parallel draft model distribution $q(\cdot|\cdot)$ and initial prefix sequence $x_{<n}$.
Return the generated sequence $y$.
Initialise $n \leftarrow t, y \leftarrow x_{<n}$.
**while** $n < T$ **do**
    Draft future token tree $\mathcal{N}$ by sampling from the parallel draft model with a single forward pass:

$$\mathcal{N} \sim (q(x|x_{<n}), q(x|x_{<n}, [\texttt{MASK}]_1), \ldots, q(x|x_{<n}, [\texttt{MASK}]_1, \ldots, [\texttt{MASK}]_k))$$

    In parallel, compute a set of logits $\mathcal{O}$ with the target model using $\mathcal{N}$ and tree attention.

$$\mathcal{O} = \text{TreeParallelDecode}(\mathcal{N}, p)$$

    $\mathcal{V} = \emptyset, u$    ▷ $u$ is the root node of $\mathcal{N}$
    **while** $u$ is not a leaf node **do**
        $\mathcal{H} = \text{Child}(u)$    ▷ $\text{Child}(u)$ returns the child nodes of $u$
        **while** $\mathcal{H}$ is not empty **do**
            Sample $r \sim U[0, 1]; s = \text{Select}(\mathcal{H}); \tilde{x}_s = \mathcal{H}(s); t = \text{TreeDepth}(s)$
            **if** $r < \min\left(1, \dfrac{p(\tilde{x}_s|x_{<n}, \mathcal{V})}{q(\tilde{x}_s|x_{<n}, \ldots, [\texttt{MASK}]_{t-1})}\right)$ **then**
                ✓ accept the draft token $x_s$ at depth $t$
                $\mathcal{V}.\text{append}(x_s); u = s; n \leftarrow n + 1$
                **break**
            **else**
                × reject the draft token $x_s$. Normalize the residual.
                $p(x|x_{<n}, \ldots, \mathcal{V}) := (p(x|x_{<n}, \ldots, \mathcal{V}) - q(x|x_{<n}, \ldots, [\texttt{MASK}]_{t-1}))_+$
                $\mathcal{H}.\text{pop}(s)$
            **end if**
        **end while**
        **if** $\mathcal{H}$ is empty **then**
            **break**
        **end if**
    **end while**
    $x_{\text{next}} \sim p(x|x_{<n}, \ldots, \mathcal{V}); n \leftarrow n + 1; \mathcal{V}.\text{append}(x_{\text{next}})$
    $y \leftarrow y + \mathcal{V}$
**end while**

---

## 5 EXPERIMENTS

This section describes the experimental settings (§5.1), including training and evaluation datasets, metrics, and involved baselines. §5.2 reports the main results for PARALLELSPEC compared with baselines. Finally, we discuss the impact of different experiment settings in §5.3.

To integrate PARALLELSPEC into Medusa, we introduce a Transformer model specialized in multi-token prediction to replace $K$ Medusa heads as the new draft model. The draft model shares the embedding layer and the language model head with the target model to minimize memory overhead, and they remain frozen to preserve the target model's output distribution. $K$ trainable $[\texttt{MASK}]$ tokens are added to the embedding layer of the draft model to facilitate parallel training. During the training, each *parallel group* is trained with a similar objective denoted in Equation 2, except that the log term now denotes the output distribution of the parallel drafter at position $k$, and we need to consider the non-mask token at the beginning of each *parallel group*:

$$\mathcal{L}_{\text{MEDUSA-Parallel}} = -\log q\left(y_{t+1}|x_{<t}\right) - \sum_{k=1}^{K} \lambda_k \log q\left(y_{t+k+1}|x_{<t}, [\texttt{MASK}]_1, \ldots, [\texttt{MASK}]_k\right). \quad (3)$$

**EAGLE** utilizes an *online* knowledge distillation method that directly regresses the last-layer hidden states at the feature level. Specifically, we denote the last-layer hidden states of the target model at $t$-th position as $f_t$, the embedding of $t$-th token as $e_t$, and the oracle token as $y_t$. EAGLE proposed to use a fully connected layer and an auto-regression head $\pi\left(\tilde{f}_t|f_{<t}, e_{<t}\right)$ as the draft model to predict the next feature that is used to decode draft tokens. The training objective of EAGLE on each token position is a linear combination of the regression loss $\mathcal{L}_{\text{reg}}$ and the cross-entropy loss $\mathcal{L}_{\text{cls}}$ between draft tokens and oracle tokens:

$$\begin{aligned}
\mathcal{L}_{\text{reg}} &= \text{SmoothL1}\left(f_{t+1}, \pi\left(\tilde{f}_t|f_{<t}, e_{<t}\right)\right), \\
\mathcal{L}_{\text{cls}} &= -\log \mu\left(y_{t+1}\right),
\end{aligned} \quad (4)$$

where $\mu$ denotes the language model head distribution conditioned on drafted feature $\tilde{f}_t$.

Integrating PARALLELSPEC into EAGLE is more intuitive, as it only requires minor modifications to turn the auto-regression head into a parallel head with the method outlined in §4.1, without extra adaptation. For a parallel group of size $K$ starting at token position $t$, the training loss of EAGLE-Parallel is the sum of EAGLE losses (defined in Equation 4) over $K$ tokens within the same group. At inference time, EAGLE integration needs an additional effort. Each drafting step uses only the embedding of [MASK] tokens with the target features of [MASK] tokens left empty, *i.e.*:

$$\tilde{f}_t, \ldots, \tilde{f}_{t+K+1} = \pi\left([f_{<t}, 0, \ldots 0]; [e_{<t}, e_{[MASK_1]}, \ldots, e_{[MASK_K]}]\right). \quad (5)$$

This is because these introduced [MASK] tokens are not in the original vocabulary of the target models; therefore, there is no way to produce target features for these [MASK] tokens. We only use the trainable embeddings of [MASK] as a signal for the drafter to predict tokens at different future time steps.

## 5.1 SETTINGS

**Datasets, Tasks, and Training.** Following the setup of SpecBench (Xia et al., 2024), we conduct evaluations on six types of text generation tasks, including MT-bench (Zheng et al., 2023) for multi-turn conversation, CNN/Daily Mail (Nallapati et al., 2016) for text summarization, Natural Questions (Karpukhin et al., 2020) for retrieval-augmented generation and question answering, WMT14 DE-EN (Bojar et al., 2014) for machine translation, GSM8K (Cobbe et al., 2021) for mathematical reasoning. As PARALLELSPEC falls into the category of speculative decoding methods that require an extra alignment stage, supervised fine-tuning (i.e., SFT) data are needed for aligning distributional similarity between the drafter and the target model. To ensure a fair comparison with baselines in this category, we follow Li et al. (2024b) to use 68,000 ShareGPT (Tay et al., 2023) conversations as training data without self-distillation. For the self-distillation setting where multi-turn conversations are distilled from the target model given the prompts from the dataset, we refer to §5.3. Due to the group-wise parallel training strategy, the training sequences for PARALLELSPEC will become longer than the ones of conventional auto-regressive drafter. Training PARALLELSPEC on 7B models takes 13 hours on 8 A100-PCIE-40GB GPUs for 40 epochs. The size of *parallel group* is set to 5, *i.e.*, the number of [MASK] tokens $k = 4$ unless stated otherwise. We refer to Appendix A.2 for details such as computing environment, other hyper-parameters, *etc*.

**Baselines.** We select seven competitive speculative decoding methods as baselines. Some of them work as plugin modules like Speculative Sampling (SpS) (Chen et al., 2023a), Prompt Lookup Decoding (PLD) (Saxena, 2023; Yang et al., 2023) and Lookahead Decoding (Fu et al., 2024), which

---

[1]While we name this model Medusa + PARALLELSPEC, we clarify that only Medusa-style loss is used in the Medusa PARALLELSPEC integration, and no more Medusa heads are utilized as drafter.

| Model | Method | Multi-turn Conversation | Translation | Summarization | Question Answering | Mathematical Reasoning | Retrieval-aug. Generation | Avg. | $\tau$ (tokens) |
|---|---|---|---|---|---|---|---|---|---|
| | | | | | Temperature = 0.0 | | | | |
| V 7B | SpS | $1.67\times_{\pm 0.04}$ | $1.13\times_{\pm 0.02}$ | $1.71\times_{\pm 0.01}$ | $1.49\times_{\pm 0.04}$ | $1.50\times_{\pm 0.03}$ | $1.67\times_{\pm 0.02}$ | $1.53\times_{\pm 0.03}$ | 2.27 |
| | PLD | $1.61\times_{\pm 0.02}$ | $1.02\times_{\pm 0.01}$ | $2.57\times_{\pm 0.02}$ | $1.14\times_{\pm 0.02}$ | $1.61\times_{\pm 0.01}$ | $1.82\times_{\pm 0.06}$ | $1.62\times_{\pm 0.01}$ | 1.75 |
| | Hydra | $2.50\times_{\pm 0.02}$ | $1.94\times_{\pm 0.02}$ | $1.89\times_{\pm 0.04}$ | $2.02\times_{\pm 0.04}$ | $2.53\times_{\pm 0.02}$ | $1.86\times_{\pm 0.07}$ | $2.13\times_{\pm 0.04}$ | 3.26 |
| | Lookahead | $1.48\times_{\pm 0.02}$ | $1.15\times_{\pm 0.02}$ | $1.36\times_{\pm 0.02}$ | $1.27\times_{\pm 0.02}$ | $1.59\times_{\pm 0.03}$ | $1.23\times_{\pm 0.03}$ | $1.35\times_{\pm 0.02}$ | 1.64 |
| | Medusa | $1.60\times_{\pm 0.01}$ | $1.39\times_{\pm 0.01}$ | $1.22\times_{\pm 0.03}$ | $1.37\times_{\pm 0.00}$ | $1.68\times_{\pm 0.01}$ | $1.20\times_{\pm 0.06}$ | $1.42\times_{\pm 0.01}$ | 2.39 |
| | +PARALLELSPEC [1] | $2.63\times_{\pm 0.02}$ | $1.97\times_{\pm 0.02}$ | $2.32\times_{\pm 0.04}$ | $2.20\times_{\pm 0.02}$ | $2.78\times_{\pm 0.04}$ | $1.98\times_{\pm 0.03}$ | $2.31\times_{\pm 0.02}$ | 3.31 |
| | Medusa† | $1.87\times_{\pm 0.01}$ | $1.56\times_{\pm 0.01}$ | $1.49\times_{\pm 0.02}$ | $1.56\times_{\pm 0.02}$ | $1.85\times_{\pm 0.04}$ | $1.42\times_{\pm 0.02}$ | $1.63\times_{\pm 0.02}$ | 2.31 |
| | EAGLE-2 | $2.68\times_{\pm 0.05}$ | $1.78\times_{\pm 0.04}$ | $2.23\times_{\pm 0.03}$ | $2.04\times_{\pm 0.04}$ | $2.69\times_{\pm 0.04}$ | $2.02\times_{\pm 0.03}$ | $2.24\times_{\pm 0.04}$ | 4.34 |
| | EAGLE | $2.57\times_{\pm 0.04}$ | $1.85\times_{\pm 0.02}$ | $2.17\times_{\pm 0.02}$ | $2.03\times_{\pm 0.04}$ | $2.57\times_{\pm 0.03}$ | $1.92\times_{\pm 0.04}$ | $2.18\times_{\pm 0.03}$ | **3.58** |
| | +PARALLELSPEC | $\mathbf{3.01}\times_{\pm 0.04}$ | $\mathbf{2.09}\times_{\pm 0.01}$ | $\mathbf{2.62}\times_{\pm 0.06}$ | $\mathbf{2.40}\times_{\pm 0.03}$ | $\mathbf{2.84}\times_{\pm 0.05}$ | $\mathbf{2.36}\times_{\pm 0.02}$ | $\mathbf{2.55}\times_{\pm 0.03}$ | 3.52 |
| L2 7B | SpS | $1.33\times_{\pm 0.03}$ | $1.25\times_{\pm 0.03}$ | $1.21\times_{\pm 0.02}$ | $1.30\times_{\pm 0.01}$ | $1.34\times_{\pm 0.02}$ | $1.43\times_{\pm 0.03}$ | $1.31\times_{\pm 0.02}$ | 1.67 |
| | PLD | $1.42\times_{\pm 0.01}$ | $1.17\times_{\pm 0.02}$ | $1.44\times_{\pm 0.02}$ | $1.07\times_{\pm 0.02}$ | $1.31\times_{\pm 0.02}$ | $1.57\times_{\pm 0.01}$ | $1.33\times_{\pm 0.01}$ | 1.42 |
| | Lookahead | $1.46\times_{\pm 0.05}$ | $1.36\times_{\pm 0.04}$ | $1.34\times_{\pm 0.04}$ | $1.32\times_{\pm 0.03}$ | $1.47\times_{\pm 0.04}$ | $1.37\times_{\pm 0.03}$ | $1.39\times_{\pm 0.04}$ | 1.60 |
| | EAGLE | $2.61\times_{\pm 0.02}$ | $2.38\times_{\pm 0.02}$ | $2.25\times_{\pm 0.02}$ | $2.30\times_{\pm 0.05}$ | $2.66\times_{\pm 0.06}$ | $2.23\times_{\pm 0.01}$ | $2.40\times_{\pm 0.02}$ | **3.55** |
| | +PARALLELSPEC | $\mathbf{2.95}\times_{\pm 0.02}$ | $\mathbf{2.67}\times_{\pm 0.01}$ | $\mathbf{2.64}\times_{\pm 0.03}$ | $\mathbf{2.76}\times_{\pm 0.04}$ | $\mathbf{2.88}\times_{\pm 0.02}$ | $\mathbf{2.52}\times_{\pm 0.03}$ | $\mathbf{2.74}\times_{\pm 0.03}$ | 3.49 |
| V 13B | SpS | $1.69\times_{\pm 0.01}$ | $1.16\times_{\pm 0.01}$ | $1.78\times_{\pm 0.00}$ | $1.45\times_{\pm 0.01}$ | $1.60\times_{\pm 0.01}$ | $1.76\times_{\pm 0.04}$ | $1.57\times_{\pm 0.01}$ | 2.18 |
| | Medusa | $2.06\times_{\pm 0.01}$ | $1.77\times_{\pm 0.02}$ | $1.66\times_{\pm 0.04}$ | $1.74\times_{\pm 0.01}$ | $2.12\times_{\pm 0.01}$ | $1.62\times_{\pm 0.05}$ | $1.84\times_{\pm 0.02}$ | 2.39 |
| | +PARALLELSPEC | $2.76\times_{\pm 0.04}$ | $2.37\times_{\pm 0.02}$ | $2.16\times_{\pm 0.02}$ | $2.33\times_{\pm 0.03}$ | $2.62\times_{\pm 0.03}$ | $2.27\times_{\pm 0.04}$ | $2.52\times_{\pm 0.05}$ | 3.34 |
| | EAGLE | $2.78\times_{\pm 0.02}$ | $2.03\times_{\pm 0.03}$ | $2.41\times_{\pm 0.02}$ | $2.11\times_{\pm 0.03}$ | $2.78\times_{\pm 0.04}$ | $2.20\times_{\pm 0.03}$ | $2.39\times_{\pm 0.02}$ | **3.64** |
| | +PARALLELSPEC | $\mathbf{3.03}\times_{\pm 0.04}$ | $\mathbf{2.30}\times_{\pm 0.03}$ | $\mathbf{2.65}\times_{\pm 0.02}$ | $\mathbf{2.36}\times_{\pm 0.03}$ | $\mathbf{3.04}\times_{\pm 0.05}$ | $\mathbf{2.46}\times_{\pm 0.04}$ | $\mathbf{2.64}\times_{\pm 0.02}$ | 3.56 |
| L2 13B | SpS | $1.38\times_{\pm 0.03}$ | $1.30\times_{\pm 0.03}$ | $1.26\times_{\pm 0.04}$ | $1.36\times_{\pm 0.03}$ | $1.41\times_{\pm 0.02}$ | $1.47\times_{\pm 0.06}$ | $1.36\times_{\pm 0.03}$ | 1.66 |
| | EAGLE | $2.80\times_{\pm 0.01}$ | $2.60\times_{\pm 0.02}$ | $2.53\times_{\pm 0.06}$ | $2.42\times_{\pm 0.04}$ | $2.85\times_{\pm 0.03}$ | $2.39\times_{\pm 0.12}$ | $2.60\times_{\pm 0.03}$ | **3.66** |
| | +PARALLELSPEC | $\mathbf{3.02}\times_{\pm 0.02}$ | $\mathbf{2.81}\times_{\pm 0.03}$ | $\mathbf{2.77}\times_{\pm 0.07}$ | $\mathbf{2.68}\times_{\pm 0.03}$ | $\mathbf{3.00}\times_{\pm 0.02}$ | $\mathbf{2.74}\times_{\pm 0.04}$ | $\mathbf{2.84}\times_{\pm 0.04}$ | 3.60 |
| | | | | | Temperature = 1.0 | | | | |
| V 7B | SpS | $1.35\times_{\pm 0.00}$ | $1.01\times_{\pm 0.00}$ | $1.39\times_{\pm 0.02}$ | $1.25\times_{\pm 0.01}$ | $1.29\times_{\pm 0.02}$ | $1.38\times_{\pm 0.05}$ | $1.28\times_{\pm 0.01}$ | 1.82 |
| | PLD | $1.56\times_{\pm 0.00}$ | $0.98\times_{\pm 0.01}$ | $2.49\times_{\pm 0.01}$ | $1.12\times_{\pm 0.01}$ | $1.56\times_{\pm 0.01}$ | $1.73\times_{\pm 0.01}$ | $1.57\times_{\pm 0.00}$ | 1.70 |
| | Lookahead | $1.43\times_{\pm 0.00}$ | $1.10\times_{\pm 0.01}$ | $1.32\times_{\pm 0.00}$ | $1.21\times_{\pm 0.01}$ | $1.53\times_{\pm 0.01}$ | $1.16\times_{\pm 0.00}$ | $1.29\times_{\pm 0.00}$ | 1.64 |
| | EAGLE | $2.10\times_{\pm 0.01}$ | $1.59\times_{\pm 0.02}$ | $1.83\times_{\pm 0.05}$ | $1.70\times_{\pm 0.02}$ | $2.04\times_{\pm 0.02}$ | $1.78\times_{\pm 0.06}$ | $1.84\times_{\pm 0.01}$ | **3.18** |
| | +PARALLELSPEC | $\mathbf{2.32}\times_{\pm 0.01}$ | $\mathbf{1.78}\times_{\pm 0.02}$ | $\mathbf{2.06}\times_{\pm 0.04}$ | $\mathbf{1.89}\times_{\pm 0.02}$ | $\mathbf{2.10}\times_{\pm 0.01}$ | $\mathbf{1.96}\times_{\pm 0.02}$ | $\mathbf{2.02}\times_{\pm 0.03}$ | 3.09 |
| L2 7B | SpS | $1.11\times_{\pm 0.01}$ | $1.07\times_{\pm 0.01}$ | $1.04\times_{\pm 0.02}$ | $1.09\times_{\pm 0.01}$ | $1.13\times_{\pm 0.01}$ | $1.15\times_{\pm 0.01}$ | $1.10\times_{\pm 0.01}$ | 1.47 |
| | EAGLE | $2.19\times_{\pm 0.02}$ | $1.92\times_{\pm 0.05}$ | $1.91\times_{\pm 0.03}$ | $1.93\times_{\pm 0.05}$ | $2.31\times_{\pm 0.05}$ | $1.87\times_{\pm 0.07}$ | $2.02\times_{\pm 0.04}$ | **3.30** |
| | +PARALLELSPEC | $\mathbf{2.47}\times_{\pm 0.03}$ | $\mathbf{2.15}\times_{\pm 0.02}$ | $\mathbf{2.08}\times_{\pm 0.04}$ | $\mathbf{2.11}\times_{\pm 0.03}$ | $\mathbf{2.42}\times_{\pm 0.01}$ | $\mathbf{2.06}\times_{\pm 0.03}$ | $\mathbf{2.22}\times_{\pm 0.02}$ | 3.25 |

Table 1: Speedup ratios and average acceptance lengths $\tau$ of different methods tested on an A100-PCIE-40GB GPU using third-party benchmark toolkit SpecBench (Xia et al., 2024). V: Vicuna-v1.3. L2: LLaMA2-Chat. We report the mean and standard deviation of speedup ratios on 3 different runs. Best metrics for each model are marked in **boldface**. † denotes additional evaluation that runs on an RTX-4090 GPU.

do not need additional training. For SpS methods, we use vicuna-68m[2] and llama-68m[3] as the drafter for Vicuna and Llama target models, respectively. SpS implementation strictly follows Spec-Bench (Xia et al., 2024) and Huggingface (Wolf et al., 2019) assisted_generation setup, where the number of draft tokens per step $\gamma$ is updated with heuristic rules. For PLD, we follow the default settings of n-gram size = 3 and number of lookup tokens = 10. For Lookahead Decoding, we use the official recommended configuration of level = 5, window size = 7, and n-gram size = 7. The remaining methods, including Medusa (Cai et al., 2024b), Hydra (Ankner et al., 2024), and EAGLE (Li et al., 2024b), that require extra training while preserving the output distributions, are the main peer works for comparison. We use their official drafter checkpoints to report results.

**Models.** We conduct experiments on the Vicuna series (7B, 13B) (Zheng et al., 2023) and the Llama-2-Chat series (7B, 13B) (Touvron et al., 2023). We chose these models as they are highly representative, and most prior methods built their drafters upon these models, allowing a fair comparison. We provide results for more recent target models in §5.3. All parallel drafters are constructed with a single Transformer layer, with hyper-parameters identical to those of layers in their target models. This results in a 202M drafter for 7B models and a 317M one for 13B models. We keep the same draft token tree structure with the selected two baseline methods in PARALLELSPEC.

**Metrics.** Similar to other speculative decoding methods, we primarily focus on the end-to-end wall-time speedup ratio compared to naive auto-regressive decoding. We also report the average acceptance length $\tau$ in each round of speculative decoding.

---

[2] https://huggingface.co/double7/vicuna-68m
[3] https://huggingface.co/JackFram/llama-68m

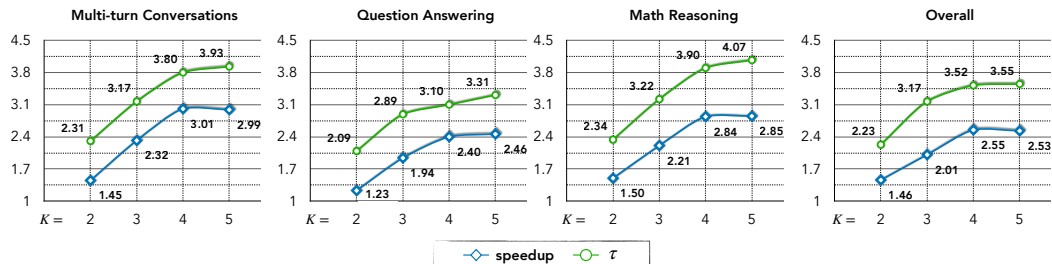

Figure 4: Ablations on speedup ratio and average acceptance length $\tau$ with respect to the number of [MASK] tokens $K$ on all three test datasets.

| Model & Method | Multi-turn Conversation | Translation | Summarization | Question Answering | Mathematical Reasoning | Retrieval-aug. Generation | Avg. | $\tau$ (tokens) |
|---|---|---|---|---|---|---|---|---|
| LLaMA 2 7B w/ EAGLE | 2.61× | 2.38× | 2.25× | 2.30× | 2.66× | 2.23× | 2.40× | 3.55 |
| +PARALLELSPEC | 2.95×$_{+13.0\%}$ | 2.67×$_{+12.2\%}$ | 2.64×$_{+17.3\%}$ | 2.76×$_{+20.0\%}$ | 2.88×$_{+8.3\%}$ | 2.52×$_{+13.0\%}$ | 2.74×$_{+14.2\%}$ | 3.49 |
| +PARALLELSPEC +self-distillation | **3.02×**$_{+15.7\%}$ | **2.77×**$_{+17.3\%}$ | **2.71×**$_{+20.4\%}$ | **2.87×**$_{+24.8\%}$ | **2.98×**$_{+12.0\%}$ | **2.56×**$_{+14.8\%}$ | **2.82×**$_{+17.5\%}$ | **3.78** |

Table 2: Ablations on alignment training data.

## 5.2 MAIN RESULTS

Table 1 gives a performance overview of PARALLELSPEC and established prior approaches on different types of text generation benchmarks. We refer to Appendix A.1 for qualitative case studies. In general, PARALLELSPEC integration brings a consistent acceleration improvement in all domains, two selected methods, and different decoding temperatures. Based on the results, we have the following key observations:

**PARALLELSPEC significantly accelerates Medusa frameworks by a large margin.** For example, integrating PARALLELSPEC into Vicuna-7B Medusa boosts the average speedup ratio from 1.42× to 2.31×, resulting in a 62.7% relative improvement. This even outperforms EAGLE, which can be attributed to the parallel drafter design. It not only increases the average acceptance tokens from 2.39 to 3.31 but also significantly reduces drafter runtime overhead. Similar improvements are also observed on Vicuna-13B Medusa, showing PARALLELSPEC is not sensitive to the target model size.

**Equipping EAGLE with PARALLELSPEC also achieves considerable speedups.** For instance, the average speedup ratio on Vicuna-7B increased from 2.18× to 2.55×; on Llama2-7B-Chat, it rose from 2.40× to 2.74×; and on LLaMA-13B-Chat, it went from 2.60× to 2.84×. It is worth noting that incorporating PARALLELSPEC into EAGLE causes a slight drop in average acceptance length. This is fully expected, as we no longer preserve the sequence dependency during the draft stage, and the parallel decoding leads to a small decline in drafting accuracy. In addition, we also conduct experiments on EAGLE-PARALLELSPEC with temperature decoding settings and observe substantial overall speedup up to 2.22×. Still, the speculative decoding efficiency at high temperature degrades compared with greedy decoding, echoing conclusions in previous studies (Xia et al., 2024).

## 5.3 ABLATION STUDIES

**Training Data.** Prior works often assume the availability of training data that aligns with the output distribution of target models, which is not usually the case. Thus, directly using ShareGPT conversations as SFT data for drafter training may suffer from the domain shift. In order to bypass this, we follow Cai et al. (2024b) to employ the self-distillation technique to build the training dataset that matches the target model. Specifically, we employ vLLM (Kwon et al., 2023) to obtain distilled multi-turn conversations by feeding the prompts from ShareGPT in a greedy decoding setting. Table 2 demonstrates that the self-distillation technique can further improve the speedup ratio, with the average acceptance tokens greatly increased to 3.78.

**Size of Parallel Group.** The size of the *parallel group* is an important hyper-parameter of our method, and it determines how many tokens the parallel drafter will predict at each step. It equals

| Model & Method | Multi-turn Conversation | Translation | Summarization | Question Answering | Mathematical Reasoning | Retrieval-aug. Generation | Avg. | $\tau$ (tokens) |
|---|---|---|---|---|---|---|---|---|
| LLaMA 3 8B w/ EAGLE | 2.66× | 2.42× | 2.33× | 2.32× | 2.75× | 2.32× | 2.47× | **3.42** |
| +PARALLELSPEC | **3.03×**$_{+13.9\%}$ | **2.63×**$_{+8.7\%}$ | **2.61×**$_{+12.0\%}$ | **2.83×**$_{+22.0\%}$ | **2.92×**$_{+6.2\%}$ | **2.45×**$_{+5.6\%}$ | **2.75×**$_{+11.3\%}$ | 3.36 |

Table 3: Ablations on recent advanced target models.

the number of [MASK] tokens $K$ plus one. To investigate its impact on the speedup performance, we conduct ablation studies to re-train parallel drafter with different $K$ and report the speedup ratio and average acceptance length in Figure 4. We notice the increase in the average acceptance length $\tau$ resulting from larger *parallel group* size is steady for small $K$ but saturates after $K = 4$, leading the speedup ratio to no longer improve beyond that point. We believe this could be attributed to the difficulty of predicting distant future tokens using parallel decoding. The benefit of increasing the *parallel group* size cannot counterbalance the overhead of predicting distant tokens, resulting in a speedup ratio sweet spot around $K = 4$.

**Advanced Target Model.** Table 3 reflects that more advanced models like LLaMA3-8B-Instruct (AI@Meta, 2024) can still benefit from the design of PARALLELSPEC. However, the relative improvements on LLaMA3-Instruct series are slightly lower than those on LLaMA2-Chat and Vicuna, possibly because of the larger misalignment between ShareGPT SFT data and LLaMA3-Instruct.

## 6 CONCLUSION

In this paper, we introduce PARALLELSPEC, a powerful speculative decoding solution that could be inserted into popular speculative decoding frameworks. It proposes to use a single lightweight model and several trainable [MASK] tokens to facilitate fast multi-token prediction as drafters, thereby mitigating the issue of drafting latency scaling linearly with the draft length. Compared with the Medusa-style multi-head multi-token draft strategy, PARALLELSPEC demonstrates significant advantages in drafting accuracy, latency, and parameter efficiency. Extensive experiments on various benchmarks and different target models demonstrate the compatibility and performance superiority of PARALLELSPEC.

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

# A  APPENDIX

## A.1  CASE STUDY

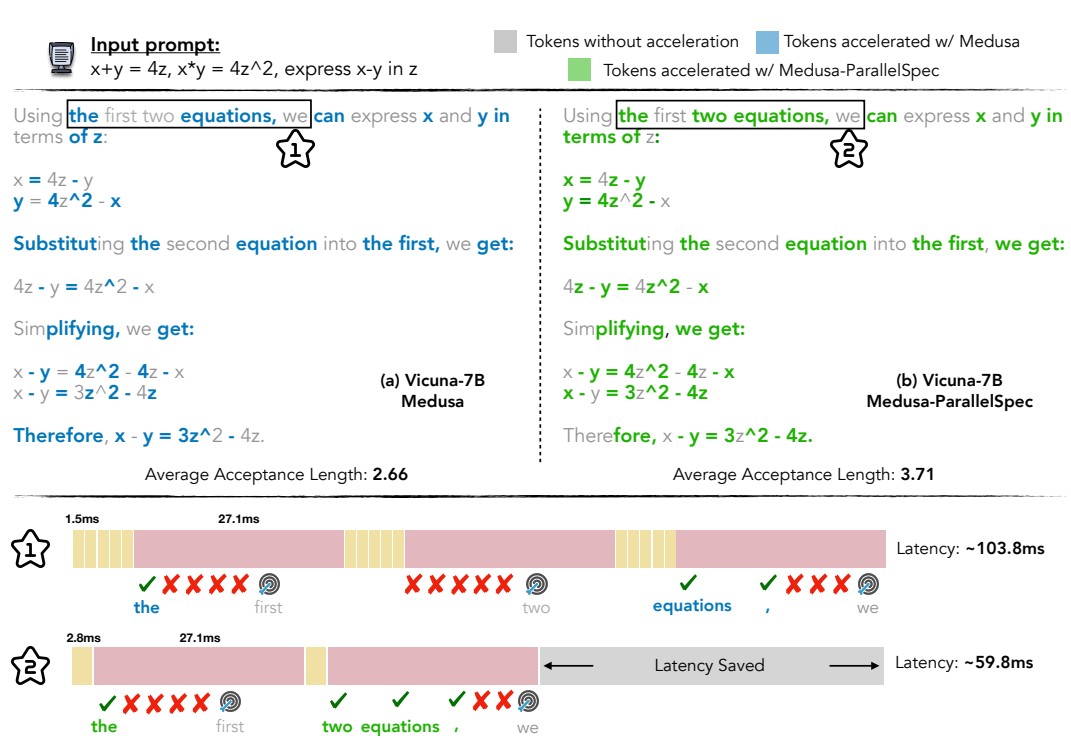

Figure 5: **Upper:** Visualization of accelerated tokens in generation from (a) Vicuna-7B Medusa and (b) Vicuna-7B Medusa-PARALLELSPEC given an input prompt from GSM8K (Cobbe et al., 2021). **Lower:** Simulated wall-time trace of two different methods generating the text in the highlighted box. We only consider the forward pass latency of draft and verification while ignoring the negligible post-processing overhead. ✔: accepted draft tokens. ✘: rejected draft tokens. ◉: tokens without speculative acceleration.

We provide an illustration of two different speculative decoding methods running on the same prompt from GSM8K in Figure 5. The side-by-side comparison indicates that Vicuna-7B Medusa equipped with PARALLELSPEC not only achieves a higher average acceptance length (3.71 vs. 2.66) but also shows a significant advantage in end-to-end latency, thanks to the one-pass decoding nature of parallel-decoding drafter. The lower part of Figure 5 even reveals that PARALLELSPEC nearly cut half of the time cost when decoding the same text span compared with the baseline method.

## A.2  EXPERIMENTAL DETAILS

| Hyper Parameter | Medusa-PARALLELSPEC | EAGLE-PARALLELSPEC |
|---|---|---|
| **Global Batch Size** | 32 | 32 |
| **Learning Rate** | $5 \times 10^{-4}$ | $5 \times 10^{-5}$ |
| **Optimizer** | AdamW ($\beta_1 = 0.9, \beta_2 = 0.999$) | AdamW($\beta_1 = 0.9, \beta_2 = 0.95$) |
| **Weight Decay** | 0.0 | 0.0 |
| **Epochs** | 20 | 40 |

Table 4: Hyper parameters of PARALLELSPEC variants for 7B models.

All training was conducted using a node with 8 NVIDIA A100-PCIE-40GB GPUs, 2 AMD EPYC 7282 CPUs, and 512GB RAM. Evaluations were conducted using one GPU of the above node. PyTorch 2.2.0 with CUDA 12.1 version was used in all experiments. To avoid the discrepancies brought by computing environment differences, we re-ran all baseline methods and our method 3 times and reported the mean speedup ratio. We list hyper-parameters used in PARALLELSPEC in Table 4. One might notice that the baseline results have around 10% differences in terms of speedup ratios compared with the original Spec-Bench (Xia et al., 2024). First, we refer you to the latest benchmark page[4] for updated results. We also appreciate your understanding that, due to budget and practical constraints, we do not have access to the same computational resources as the Spec-Bench team. Therefore, we were not able to reproduce their speedup results. However, since all the improvements in this paper were obtained using the same hardware environment, our comparisons are relatively fair. We also notice that in the Spec-Bench paper, Table 5 and Table 6 also reported entirely different sets of speedup ratios when the only difference is the GPU used for benchmarking, indicating the hardware specifications are one of the major factors that impact reported speedup.

### A.3 TOKEN TREE VERIFICATION

Initially proposed in SpecInfer (Miao et al., 2024), and also explored in several follow-up works (Cai et al., 2024b; Li et al., 2024b; Spector & Re, 2023), tree-based structure for token verification is proved to be useful. Following existing studies, PARALLELSPEC leverages tree attention to realize this process. Specifically, to guarantee that each token only accesses its predecessors, we use an attention mask that exclusively permits attention flow from the current token back to its antecedent tokens. The positional indices for positional encoding are adjusted in line with this structure. A conceptual view of this process is visualized in Figure 6, with the draft tree structure in the figure being adopted in all experiments. As the main contribution of this paper is not a novel token tree verification strategy, we adopt the same static token trees used in Medusa-1 (Cai et al., 2024b) and EAGLE (Li et al., 2024b). Starting from "Root", every node is expanded with $k$ tokens with top-$k$ highest probabilities. $k$ is designed based on manually crafted rules, and is dynamically changing as the tree depth grows.

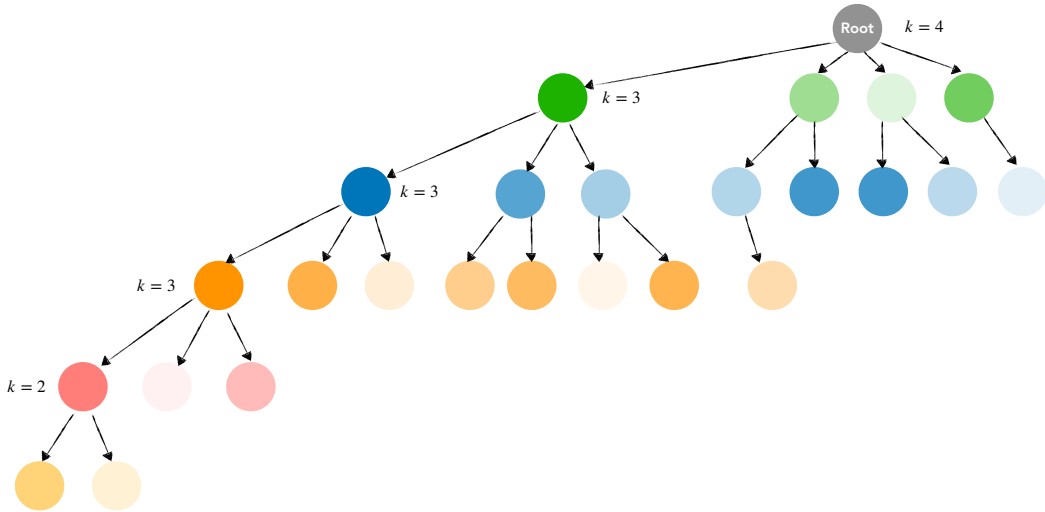

Figure 6: An example of the tree-structured verification process. Each circle represents a token, and the shading of the color indicates the probability of each token in the distribution. Tokens with the highest probability are selected and gradually expanded with dynamically designed $k$.

---

[4]https://github.com/hemingkx/Spec-Bench/blob/main/Leaderboard.md

