# OpenReview forum: "ParallelSpec: Parallel Drafter for Efficient Speculative Decoding"
_ICLR.cc/2025/Conference — Submitted to ICLR 2025_

### Official Review · Reviewer_YUuo · 2024-10-17

**Soundness:** 3
**Presentation:** 3
**Contribution:** 4
**Rating:** 8
**Confidence:** 4

**Summary:**

This paper introduces a new technique, ParallelSpec, to advance the topic of speculative decoding. The proposed method addresses the efficiency barrier in autoregressive decoding within the draft-then-verify paradigm by adding new [MASK] tokens and predicting the next $K$ tokens in a single draft model call. ParallelSpec is orthogonal to methods that require training a draft model through distillation with the target model. To improve the ability to predict multiple subsequent tokens, the paper proposes "group-wise parallel training," where groups of [MASK_1] ... [MASK_K] are inserted after each token in the training corpus, with predicting the correct token at the corresponding position as the training objective. Extensive experiments evaluate the effectiveness of ParallelSpec's design. ParallelSpec is implemented on Llama-2-chat-7B/13B and Vicuna-7B/13B, trained on top of Medusa and EAGLE, and evaluated on SpecBench, where it is compared to popular baselines for speculative decoding. ParallelSpec outperforms all baselines on the tested data, achieving greater wall-time speedup ratios. The paper also includes comprehensive ablation studies, investigating the effects of different $K$ values and self-distillation. Overall, ParallelSpec is an effective technique that can be combined with most speculative decoding algorithms to improve efficiency.

**Strengths:**

This paper makes significant contributions to the topic of speculative decoding, offering a technique that can enhance speedup ratios. There are several strengths of this paper (which I personally find very inspiring), and the most important ones are listed below:

1. Clear presentation. The paper clearly outlines the background for developing this technique, and the introduction to related works is distinct and comprehensive. Figures and tables are presented well, making them easy for readers to understand. Important hyperparameters and experimental settings are clearly stated. The way experimental results are presented is highly readable.

2. Simple but effective method. The idea of transforming the autoregressive decoding process of the draft model into parallel decoding is brilliant, and the implementation of group-wise parallel training is simple, clear, and effective. The adaptations to existing methods are minor, and the proposed method can be integrated with most prior work.

3. Strong performance. The experimental results show a substantial speedup gain when ParallelSpec is applied. The proposed method outperforms state-of-the-art methods without significant training costs.

4. Well-designed experiments. ParallelSpec is compared with popular baselines in this field using mainstream benchmark datasets. The ablation studies focus on key components of the system's design.

**Weaknesses:**

Although this paper makes valuable contributions, there are certain flaws that prevent it from receiving a higher rating. The weaknesses are listed as follows, and the overall rating could be improved if these issues are addressed:

1. Unfair competition between standalone Medusa and Medusa+ParallelSpec. In the original Medusa setup, the Medusa heads are organized in an MLP-like architecture. However, when Medusa is adapted with ParallelSpec, the draft model is a single-layer Transformer model (lines 266-267). To what extent is the speedup gain due to the change in draft model architecture, as opposed to predicting draft tokens in parallel? A baseline should be provided where the only alteration is the draft model architecture. Otherwise, the modified setting should be given a different name, as the current name might create confusion regarding the draft model architecture.

2. Unclear presentation of how EAGLE is adapted with ParallelSpec. The description in lines 355-357 is insufficient. It would greatly improve clarity to provide a more detailed explanation of the training and inference process when adapting EAGLE, even in the Appendix if space in the main text is limited. This weakness is further elaborated in Question Q1.

**Questions:**

Here are my questions and some suggestions for this paper:

1. EAGLE adaptation. In the standalone EAGLE algorithm, assuming the input embeddings are $e_{< t}$, the first step of drafting proceeds as $\tilde f_{t} = \pi (f_{<t}; e_{<t})$, and the second step goes as $\tilde f_{t+1} = \pi([f_{<t}, \tilde f_{t}]; [e_{<t}, \tilde e_t])$, where $[\cdot, \cdot]$ represents concatenation, and all other notations follow this paper. Given that the hidden states $f$ are also generated autoregressively, how is this handled in ParallelSpec? In my understanding, the draft model should forward once to generate all the draft tokens, i.e., $\tilde f_t, \dots, \tilde f_{t+K+1} = \pi ([f_{<t}, ?\dots ?]; [e_{<t}, {[MASK_1]}, \dots, {[MASK_K]}])$. What should be placed in the question marks, or is my understanding of EAGLE+ParallelSpec's drafting process incorrect?

2. Inconsistency results with official SpecBench. The baseline results of Medusa and EAGLE in Table 1 seems to be lower than the official SpecBench [1] results. The speedup of Medusa on Vicuna-7B is 1.42 in Table 1 while the official SpecBench gives 1.78, and speedup of EAGLE on Vicuna-7B in Table 1 is 2.18 while the official result is 2.29. Could you provide an explanation on the reason why the speedup gap occurs?

3. The organization of Table 2. The upper and lower halves of Table 2 should be presented in separate tables. They cover different topics: the upper half focuses on self-distillation, while the lower half discusses Llama-3.

4. Compatibility with dynamic draft trees. I am curious about the speedup gain of ParallelSpec when applied to dynamic draft trees, such as in EAGLE-2. Is this method compatible with dynamic trees, and would its strong performance persist? This is just a question for discussion, and the authors are not required to conduct such experiments during the review process.

[1] https://github.com/hemingkx/Spec-Bench/blob/main/Leaderboard.md#vicuna-7b-v13

---

> ### Comment · Reviewer_YUuo · 2024-11-20
> **Reminder for Authors' response**
>
> This is a reminder for the authors to provide a response for the reviews above. There are various weaknesses and questions to be addressed and your response determines whether a raise of the overall score can be conducted. I really like this paper and I would like to hear some answers from the authors.

---

> ### Author Response · Authors · 2024-11-22
> **Discussion (Part 1)**
>
> Dear Reviewer YUuo,
>
> We are grateful for your positive recognition and continuous interest in our work. We would be happy to address your concerns as below.
>
> > (W1) Competition between Medusa and Medusa+ParallelSpec is not fair
>
> As you mentioned, the improvements our method brings to Medusa with the new drafter structure and the introduction of parallel drafting may obscure the actual performance gains achieved solely through parallel drafting.
>
> However, as Reviewer bvJa pointed out, the Medusa drafting stage could be considered a parallel decoding stage to some extent, as there is no information exchange between the different language model heads when guessing future tokens. Therefore, we think simply adding parallel drafting to Medusa can not be seen as a significant scientific contribution since the only thing needed is to parallelize the inference of multiple language model heads.
>
> We agree that Medusa-ParallelSpec creates confusion that it still preserves the original multi-head drafter design. The difference we would like to highlight is that Medusa-ParallelSpec is trained only with Medusa loss, i.e. token-level classification loss over the vocabulary, while EAGLE-ParallelSpec is trained with EAGLE-specific feature-level regression loss together with classification loss. We will have another name on Medusa-ParallelSpec in the next version to avoid such confusion.
>
> ---
>
> > (W2 & Q1) EAGLE adaptation with ParallelSpec is not detailed.
>
> Good question! As you described, for the original EAGLE drafting, the features are autoregressively regressed with the input of last-step embedding and features from target models (i.e., target features). In the design of EAGLE-ParallelSpec, each drafting step uses only the embedding of [MASK] tokens with the target features of [MASK] tokens left empty, i.e. $\tilde{f}\_t, \ldots, \tilde{f}\_{t+K+1}=\pi\left(\left[f\_{<t}, 0, \ldots 0\right]; \left[e_{<t},\left[M A S K\_1\right], \ldots,\left[M A S K\_K\right]\right]\right)$.
>
> The reason behind that is these introduced [MASK] tokens are not in the original vocabulary of the target models; therefore, there is no way to produce meaningful target features for these [MASK] tokens. We only use the trainable embeddings of [MASK] as a signal for the drafter to predict tokens at different future time steps.
>
> We apologize for not describing the inference-time adaptation in the paper, and we think this is crucial for readers to understand EAGLE-Parallel.
>
> ---
>
> > (Q2) Inconsistent baseline results with Spec-Bench.
>
> We are also aware of the differences between baseline methods reported in ParallelSpec and Spec-Bench. Reviewer bvJa also raised similar concerns. Here are some efforts we made during the discussion stage:
>
> 1. We reached out to the Spec-Bench team for their hardware specifications when reporting results in the Spec-Bench paper. We confirmed that their team used the A100 SXM4 version to perform the evaluation, and our team used the A100 PCIE version. SXM4 version has a significantly higher rated thermal power consumption (400W v.s. 250W in the PCIE version). Also, SXM4 has a different memory bandwidth. We re-examined the evaluation log and confirmed that in our evaluation environment described in L810-L812, the Spec-Bench toolkit works as expected to report faithful results in Table 1.
>
> 2. We re-ran the Medusa method on RTX-4090 GPU and reported the results in the revised submission (L330). ParallelSpec still brought significant relative improvement in terms of speedup ratio when evaluating with a different GPU.
>
> 3. We appreciate your understanding that, due to budget and practical constraints, we do not have access to the same computational resources as the Spec-Bench team. This challenge is a common issue in the SPD community when reporting speedup results, as the essence of speculative decoding is to exploit redundant computation intensity in hardware accelerators.
>
> 4. However, since all the improvements in this paper were obtained using the same hardware environment, our comparisons are relatively fair. We also notice that in the Spec-Bench paper, Table 5 and Table 6 also reported entirely different sets of speedup ratios when the only difference is the GPU used for benchmarking, indicating the hardware specifications are one of the major factors that impact reported speedup. We add explanations in L869 to discuss this issue.
>
> ---
>
> > (Q3) Organization of Table 2.
>
> Thanks for suggesting this. We also think separating the original Table 2 will improve the clarity of our paper's presentation. Please see pages 9 and 10 for the revised table.

---

> ### Author Response · Authors · 2024-11-22
> **Discussion (Part 2)**
>
> > (Q4) Compatibility with dynamic draft trees.
>
> Thank you for bringing this up. Although the main contribution of this paper is not the token tree verification strategy—our token tree, as mentioned in Appendix A.3, is the same as those used in EAGLE and Medusa—we believe that our approach is fully compatible with the confidence-guided dynamic token tree proposed in EAGLE-2, with no conflicts between the two. We appreciate your understanding that combining ParallelSpec with the dynamic token tree is not feasible during the discussion phase due to the extensive code modifications required. That said, we are glad to report the results of EAGLE-2 on Spec-Bench in Table 1 in the revised version.
>
> We sincerely appreciate your comments, which helped improve our paper, and we hope our responses above helped to address your concerns. Please feel free to leave us more comments if you have any questions, and we are always happy to engage in discussions.

---

> > ### Comment · Reviewer_YUuo · 2024-11-22
> > **Response to Authors**
> >
> > Thanks for your response, which answers my questions mostly. However, there are some remaining questions and issues I would like to ask.
> >
> > W1: I understand that the original Medusa predicts the next k tokens independently by using multiple MLPs. However, replacing the MLPs to a transformer layers is a large modification to the original method, and this should be highlighted in the paper. The name "Medusa-ParallelSpec" should be changed in the revised version, or there should be a footnote or something highlighting this issue out.
> >
> > W2 & Q1: Thanks for making this clear. Such description should be integrated into the revised manuscript. I highly suggest to include the equation, too.
> >
> > Q4: Thanks for providing the EAGLE-2 baseline and answer my question. Could you explain the reason why the $\tau$ provided in EAGLE-2 does not align with the original result in the EAGLE-2's paper? I think that although hardware difference would result in a discrepancy in the end-to-end speed up due to multiple reasons, but the average accepted length should remain similar, as the compute is expected to be same. What are the hyperparameters used in the EAGLE-2's experiment?
> >
> > Overall, thanks your detailed response. As long as the issues above are solved and updated in the manuscript, I would consider raising my score.

---

> > > ### Author Response · Authors · 2024-11-22
> > >
> > > Thanks for the support. Here are the revisions we made and responses to your follow-up questions.
> > >
> > > W1: Yes, we totally agree with your opinion. While currently, L317-L320 already describes the difference between Medusa-ParallelSpec and Medusa, we think the name itself may still confuse. We add a footnote in Table 1 to further clarify this large modification to the Medusa method. (L377)
> > >
> > > W2 & Q1: We have incorporated the description and the equation in the revision (L346-L354). Thanks again for improving the clarity of the manuscript.
> > >
> > > Q4: We believe the difference in average accepted length with the EAGLE-2 paper-reported numbers is due to EAGLE-2 reporting the metrics on a different set of data. For example, Alpaca in EAGLE-2 evaluation is not covered by Spec-Bench. We refer you to the latest Spec-Bench leaderboard page (https://github.com/hemingkx/Spec-Bench/blob/main/Leaderboard.md), where you can see EAGLE-2 results on RTX-3090 GPU. Our reported average accepted length (4.34) is nearly identical to the Spec-Bench team reported (4.36).
> > >
> > > We hope these answers could address your concerns!

---

> > > > ### Comment · Reviewer_YUuo · 2024-11-22
> > > > **Response to Authors**
> > > >
> > > > Thanks for your clarification and swiftly update. I will increase my overall rating to 8.

---

> > > > > ### Author Response · Authors · 2024-11-22
> > > > >
> > > > > Thanks again for your positive feedback and for increasing the overall rating! We appreciate your efforts in reviewing this paper to help put it in better shape.

---

### Official Review · Reviewer_y5aG · 2024-10-17

**Soundness:** 2
**Presentation:** 2
**Contribution:** 2
**Rating:** 5
**Confidence:** 4

**Summary:**

To deal with the speculation overhead from the auto-regressive draft model in speculative decoding, the paper proposes a parallel decoding approach for speculation by adding additional mask tokens in the vocabulary. Within a single draft model forward step, several future tokens can be speculated in parallel with the [mask] tokens used as temporary input tokens. The parallel speculated tokens are then fed into the target model for verification.  Empirical evaluations show that, assisted with this parallel decoding scheme proposed by the paper, current state-of-the-art speculative decoding frameworks, including EAGLE and Medusa, can further achieve a higher speed-up ratio due to the reduction in the speculation overhead while preserving a similar speculation accuracy.

**Strengths:**

1. The problem is well-motivated as the speculation overhead can indeed play a crucial role in the end-to-end speed-up ratio brought by speculative decoding.
2. The paper's empirical evaluations show promising results for the proposed ParallelSpec approach, where the EAGLE+ParallelSoec and Medusa+ParallelSpec can consistently achieve a better speed-up ratio than the base version.
3. The presentation is clear and illustrative.

**Weaknesses:**

1. The concept of parallel decoding is not new. In fact, [1] has proposed a similar idea based on a parallel-decoded drafter and a target model as a verifier for speculative decoding. Additionally, Medusa itself is a parallel decoded speculation approach. Compared with these existing works, this paper mainly differentiates in its draft model architecture design that leverages the [MASK] tokens as a placeholder to predict future tokens within one model forward pass. However, this design is similar to the approach proposed in [2].

2. Though the empirical evaluations in the paper demonstrate promising end-to-end speed-up results on Vicuna-7B/13B and LLaMA-2-7B/13B, the evaluations on a larger model (e.g., LLaMA-2-70B) are missing. I guess this is due to the speculation overhead for a larger target model being less dominant as the theoretical improvement of minimizing speculation overhead is $\frac{\text{Latency(Verification)+Latency(Speculation)}}{Latency(Verification)}$ assuming that the speculation accuracy is the same. However, the speculation accuracy and overhead are usually a trade-off, so achieving this theoretical speed-up in practice is hard.

3. If I understand correctly, even using the same dataset, the way you trained your draft model results in a larger processed dataset due to the existence of parallel groups, making the training of your drafter less efficient than EAGLE.

4. Minor: as the EAGLE results in the SpecBench are a bit outdated, it would be interesting to compare your methods against the latest version (EAGLE-2).



**references**

[1]. Stern, Mitchell, Noam Shazeer, and Jakob Uszkoreit. "Blockwise parallel decoding for deep autoregressive models." Advances in Neural Information Processing Systems 31 (2018).

[2]. Xia, Heming, et al. "Speculative decoding: Lossless speedup of autoregressive translation." (2022).

**Questions:**

1. **Technical**-wise, how is your work different from [1] and [2] given that they both are parallel speculation frameworks and [2] even has the same [MASK] token design?
2. **Empirical**-wise, it is important to have evaluation results for a larger model (LLaMA-2-70B) where the inference speed-up is more crucial. Although the current results can show the potential of your draft model design when scaled to a larger model, it would be great if your method could still consistently achieve the best speed-up ratio on a 70B model. If time allows, it is more promising if you can compare it with EAGLE-2.

Minor issue: The wall time trace diagram (right) in Figure 1 is a bit misleading. The latency saving shown in the diagram is only guaranteed if the parallel decoded draft model has the same speculation accuracy as the auto-regressive decoded draft model. So, I would suggest adding this assumption in the caption and the paragraph that mentions the figure (the second paragraph in the introduction section).



**references**

[1]. Stern, Mitchell, Noam Shazeer, and Jakob Uszkoreit. "Blockwise parallel decoding for deep autoregressive models." Advances in Neural Information Processing Systems 31 (2018).

[2]. Xia, Heming, et al. "Speculative decoding: Lossless speedup of autoregressive translation." (2022).

---

> ### Author Response · Authors · 2024-11-22
>
> Dear Reviewer y5aG,
>
> Thank you for your insignificant comments. We believe the submission could greatly benefit from them. We would like to address your concerns below.
>
> > Parallel Decoding is not novel. How was our work different from Stern et al. and Xia et al.?
>
> We appreciate your comments on the novelty issue, and Reviewer bvJa raised similar issues. Medusa and Stern et al. initiated the use of multiple language model heads for multi-token prediction, inspiring recent efforts such as “Better & Faster Large Language Models via Multi-token Prediction.” ParallelSpec, on the other hand, adopts a compact transformer model as the drafter, using [MASK] tokens at different positions to guide predictions.
>
> Xia et al. proposed SpecDec earlier in 2022. It has not leveraged significant advancements in the speculative decoding community in recent years, such as speculative sampling, token tree verification from SpecInfer, and feature-level regression in EAGLE. Its drafter and target model were still based on an encoder-decoder architecture. Most importantly, it relaxed the verification criteria, which prevented it from achieving lossless speedup. This is reflected in the evaluation, where BLEU scores are inconsistent with naive greedy decoding. **We believe that ParallelSpec is one of the works that achieve parallel decoding using a standalone trainable drafter while maintaining lossless speedup.**
>
> ----
> > Missing results on larger target models, e.g. LLaMA-2-70B.
>
> Thanks for raising the issue regarding larger target models. While we are unable to provide results for ParallelSpec on a 70B model during the discussion phase due to resource and time limitations, we would like to clarify that the issue you mentioned is likely common across most draft-then-verify frameworks.
>
> Specifically, when the target models grow to 70B, it is customary to scale up the drafter accordingly. For instance, in the speculative sampling framework, a LLaMA-2-7B drafter would generally be a better fit for a LLaMA-2-70B target model than a LLaMA-68M drafter considering the trade-off between accuracy and overhead. The same principle applies to parallel speculation frameworks: for a 70B target model, the drafter would typically be scaled up to 1B, a single transformer layer of LLaMA-2-70B. Autoregressively forwarding a 1B drafter still incurs a non-negligible computational cost.
>
> ---
>
> > Training ParallelSpec is less efficient than EAGLE.
>
> We must acknowledge that, due to the special training strategy, the training of the parallel drafter is not as fast as that of the autoregressive drafter. In group-wise parallel training, each dynamically adjusted output token embedding alongside multiple [MASK] embeddings must simultaneously be aligned with several future token embeddings, unlike autoregressive training, which only focuses on the next token. This extends the training sequence length and results in more training FLOPs. As a reference, training EAGLE for the same number of epochs on an 8xA100 PCIe 40GB setup took around 10 hours, compared to 13 hours for ParallelSpec. We will explicitly point out this limitation of ParallelSpec in L387 of the revised version.
>
> ---
>
> > Updated EAGLE-2 results.
>
> We are glad to report the results of EAGLE-2 on Spec-Bench in Table 1 in the revised version L330. We also kindly refer you to the latest Spec-Bench Leaderboard (https://github.com/hemingkx/Spec-Bench/blob/main/Leaderboard.md) for EAGLE-2 results on Spec-Bench hardware.
>
> ---
> > Misleading Figure 1 presentation about wall time trace.
>
> We revised the caption to clarify that the saved latency could only be achieved if drafting accuracy is the same for both drafting styles given the prefix sequence.
>
> ---
>
> We sincerely thank you again for your efforts in reviewing which helped a lot in improving our paper. We hope our responses above could address your concern and please feel free to leave any comments if you have. We are always happy and open to discussions.

---

> > ### Comment · Reviewer_y5aG · 2024-11-24
> >
> > Thank you for the clarification and detailed explanation.
> >
> > Novelty-wise, I am still not convinced that your framework significantly differs from previous works. Indeed, Xia et al. introduces two different versions of their verification methods where one can guarantee the exact same output while the other one is relaxing verification for a higher speed-up ratio. In their table, they report both. Additionally, their drafter is a stand-alone model that requires a similar block-wise training scheme.
> >
> > Empirically, I would thank the author for updating the results for EAGLE-2. However, given the limitation on the novelty of the proposed methodology, a more comprehensive empirical evaluation should be conducted to meet the venue's acceptance criteria. I would be happy to recommend accepting the paper if 1(major). The methodology is verified on a larger model (if LLaMA-70B is too much effort, Vicuna-33B can also show some evidence) 2(minor). Restricting the training FLOPs to be similar to the baselines (e.g., EAGLE) to make it a more fair comparison.
> >
> > I have raised my score in recognition of the author's detailed response and explanation. However, I think the contribution in its current form doesn't meet the conference acceptance threshold.

---

> > > ### Author Response · Authors · 2024-11-25
> > >
> > > Thank you for your continued interest in our work! We’re grateful for your recognition of our response and understand your two main concerns: scalability to larger models (33B/70B) and training efficiency, which currently affect your view of its contribution. After reviewing the remaining time in the discussion stage, we’d like to provide the following response: for the larger target model experiments, we will update the paper with 33B model results and ablation studies under the same training FLOPs in its final version, regardless of the paper’s acceptance. We found that it is not feasible to complete this using our existing resources in the available timeframe. We understand this may cause frustration and fully respect your current decision.

---

### Official Review · Reviewer_bvJa · 2024-10-19

**Soundness:** 2
**Presentation:** 3
**Contribution:** 2
**Rating:** 5
**Confidence:** 4

**Summary:**

This paper proposes a new drafter architecture for speculative decoding methods, which drafts several tokens in parallel rather than auto-regressively. It suggests a process for training these drafter models, shows how to incorporate it into several state of the art speculative decoding methods such as Medusa and EAGLE, and shows that it manages to achieve around 37-62% additional speedup for Medusa, and 9-17% additional speedups for EAGLE.

**Strengths:**

**Originality:** While the idea of parallel drafting in speculative decoding was already explored, this paper is the first time the effectiveness of this method is demonstrated with a separate drafter.

**Clarity**: Overall the method is simple, elegant, and clearly presented.

**Significance**: Speculative decoding is a widely used technique for accelerating language model inference, and speeding up state of the art speculative decoding methods is an important problem with clear implications for both research and industry. The demonstrated speed-up of ~15% on top of EAGLE is notable, highlighting the practical value of this work.

**Weaknesses:**

While accelerating speculative decoding methods is an important problem, and the ~15% speedup over existing methods is commendable, I believe the paper should be rejected for the following reasons: (1) the method is not novel enough as it is very similar to ideas already present in existing works (2) the results are not overly convincing and their magnitude is generally far below the 62% reported in the abstract. (3) There are some reproducibility issues, though I think these can be easily clarified by the authors in the follow up discussions.

**Novelty**: The core idea of drafting tokens in parallel instead of autoregressively was already presented by Stern et al (“Blockwise Parallel Decoding for Deep Autoregressive Models”), Monea et al (“PaSS: Parallel Speculative Sampling”) and to some extent Medusa, and the architecture presented in the paper closely resembles Monea et al, though their work was limited to the case where the same target model also generates the look-ahead drafts. This paper, while incrementally different, applies the concept with a separate drafter.

**Reproducibility**: The paper lacks sufficient details regarding the construction of the token tree. The main text refers to Appendix A.3, which only states that each node is expanded with “top-k highest probabilities”, where k is based on “manually crafted rules”. Without more specific details for how k is chosen, this process cannot be fully reproduced. Also, figure 6 may be unintentionally misleading, as from the text description the number of children of each node on a given depth should be the same (the “k” chosen for that depth). To enhance reproducibility, I suggest including a more detailed explanation in Appendix A.3 of how k is chosen for each depth level, and perhaps including pseudo-code.

**Clarity**: There are some mistakes and inaccuracies, mainly in “algorithm 1” which I believe should be revised. I’ll outline these in the Questions/Suggestions section. These issues can likely be addressed before publications, so they did not impact my score.

**Experimental results**:
* A speed-up of ~15% over a state of the art method like EAGLE is notable, and could offer important inference improvements. However, there is some discrepancy between the results presented here and the ones presented in the SpecBench paper.
* Most importantly, for Medusa (which this paper claims to gain significant speed-ups of 30-60% over), the results presented in table 1 show significantly worse speed-ups than those in table 6 from the SpecBench paper (e.g. on 7B, SpecBench reports 2.42x while ParallelSpec paper reports 1.42x, on 13B SpecBench reports 2.16x while ParallelSpec paper reports 1.84x). I think it’s important to clarify these differences, as the 62% additional speedup reported in the abstract is the one achieved on top of these sub-par Medusa results.
* The EAGLE reported speedups are also a bit lower than in the SpecBench paper, but by a smaller magnitude (e.g. for 7B, 2.39x in SpecBench paper vs 2.18x in ParallelSpec).Still, this ~10% difference is significant since ParallelSpec only reports an improvement of 15% over EAGLE.
* The 62% speedup claim in the abstract is based on results from the Medusa 7B baseline, which demonstrates particularly subpar performance compared to SpecBench (and according to table 1, seems to be slower than even vanilla SpS or lookahead methods).

**Questions:**

**Concrete action items for the experimental results:**
1. I think it would be more accurate to temper the claim in the abstract, as the speedup for other baselines is around 9%-17% which is still notable.
2. Why are the reported speedup results for Medusa and EAGLE differ from those in the SpecBench paper, especially for Medusa where the difference is especially significant? I think it would strengthen the paper to add an explanation for this discrepancy.

**List of more minor things that could be improved in the paper:**
1. Figure 1 left - I think the arrows could be bolded or colored to clarify the distinction between “Auto-regressive Drafting” and “Parallel Drafting”. Currently the difference is a bit hard to spot.
2. End of intro section - since relative improvement (62.7%) is mentioned for Medusa, it should also be mentioned for Eagle (where I think it is 9-17%).
3. Related Work / Parallel Decoding - I believe "Blockwise Parallel Decoding for Deep Autoregressive Models" can also be mentioned here and possibly Medusa as well
4. Background / Speculative Decoding Procedures -  inconsistent notation for $x_k$. “Drafted token sequence $x_1,...,x_i$ where $x_{t+i} \sim q(y_{t+i})$” should be “Where $x_k \sim q(y_{t+k})$”.
5. I found algorithm 1 a bit hard to follow and suggest a few corrections below. However, since AFAIU this algorithm is simply standard tree speculative decoding as presented e.g. in SpecInfer, I believe it can be omitted or moved to the appendix if needed.
6. Experiments / Baselines - For SpS, it is not clear which gamma was chosen (how many draft tokens are generated on each iteration). Was gamma optimized to maximize speed-up or was it chosen based on some other criteria?

**Algorithm 1 issues:**
1. “Given … initial prefix sequence $x_{<n}$” should be $x_{<t}$, otherwise the line “Initialise n ← t” doesn’t make sense (since n can’t be initialized after it’s already used).
2. As written above, “Draft future token tree N” the details of how this tree is generated are not precisely written in the paper.
3. “O = TreeParallelDecode(N, p)”: O is set but never used. Instead, this should say something like - compute all logits p(x|x<n,{all paths from root to any node}) with the target model using N and tree attention.
4. “$\tilde{x}_s=H(s)$”: this notation is not clear, as H is a set of tree nodes and s is a node in the tree selected from $H$. I understand from the context $\tilde{x}_s$ is the token corresponding to node s in the tree but it should be made clearer.
5. I understand the notation $q(x|x_{<n},...,MASK_t)$ (since the … denotes the other mask tokens), but why $p(x|x_{<n},...,V)$? Why not just $p(x|x_{<n},V)$?
6. Normalize the residual line has an extra “+” at the end of the line.
7. The term $x_{<n}$ uses the n that keeps growing (initialized to t and then grows until it’s T or above). However, x never changes and it’s always the initial prefix of only t tokens. Should probably be changed to $x_{<t}$ everywhere.

---

> ### Author Response · Authors · 2024-11-22
> **Discussion (Part 1)**
>
> Dear Reviewer bvJa,
>
> Thank you for the constructive feedback, especially the detailed correction in Algorithm 1! We would like to introduce our response to your concerns and the status of the action items.
>
> > (W1) Novelty Issue
>
> We would like to address your concerns regarding the novelty issue compared with Monea et al， PaSS and Medusa. We view these works as significant contributions to the parallel decoding field. Medusa and Stern et al. initiated the use of multiple language model heads for multi-token prediction, inspiring recent efforts such as “Better & Faster Large Language Models via Multi-token Prediction.” ParallelSpec, on the other hand, adopts a compact transformer model as the drafter, using [MASK] tokens at different positions to guide predictions.
>
> We recognize that this approach may appear similar to PaSS. However, we believe that depending solely on several trainable lookahead tokens without other trainable parameters, as in PaSS, makes it difficult to develop a drafter that is both accurate and efficient. As you noted, our design employs a separate drafter for parallel drafting, supported by a sophisticated training strategy to ensure high training efficiency.
>
>
> > (W2) Reproducibility issue regarding the draft token tree
>
> We apologize for not clearly stating the process of constructing the token tree used in ParallelSpec. **In fact, in all experiments of ParallelSpec, we adopt the same static token trees used in Medusa-1 and EAGLE**, with the structure illustrated in Figure 6. The pseudo-code for the structure is listed below:
>
> ```python
> mc_sim_7b_63 = [[0],[1],[2],[3],[0,0],[0,1],[0,2],[1,0],[1,1],[2,0],[2,1],[3,0]
>                ,[0,0,0],[0,0,1],[0,0,2],[0,1,0],[0,1,1],[0,2,0],[0,2,1],[1,0,0],
>                [0,0,0,0],[0,0,0,1],[0,0,0,2],[0,0,0,0,0],[0,0,0,0,1]]  # depth of 5. in total 25 nodes waiting for verification
>
> ```
>
> We think the primary contribution of this paper does not lie in proposing a novel token tree verification strategy; therefore, using the same token tree ensures that the relative improvements can be fairly attributed to the parallel drafter design. We would also like to report additional results on EAGLE-2 in L330 of the revised version, where we can see even a context-dependent dynamic draft tree provided marginal improvements on Spec-Bench. We revised Appendix A.3 to describe the token tree used explicitly.
>
> > (W3) Presentation issues mainly in Algorithm 1
>
> We sincerely appreciate your meticulous review of the algorithm. To ensure Algorithm 1 is presented in its optimal state, we will carefully cross-check the notational issues in Algorithm 1 according to your suggestions after the discussion phase concludes. We are happy to see it did not impact your ratings with our paper.
>
> > (W4) Explanation for baseline performance differences with Spec-Bench
>
> We would like to first address your concerns about the significantly worse performance of Medusa baseline compared with the metrics reported in the Spec-Bench paper.
>
> 1. It is worth noting that the Spec-Bench team has updated their leaderboard (https://github.com/hemingkx/Spec-Bench/blob/main/Leaderboard.md) to reflect the corrected speedup ratios on both A100 SXM4 80GB GPU and RTX 3090 24GB GPU. **The corrected average speedup for Medusa-vicuda-7b-v1.3 is 1.78$\times$ instead of 2.42$\times$ reported in the paper.**
>
> 2. We reached out to the Spec-Bench team for their hardware specifications when reporting results in the Spec-Bench paper. We confirmed that their team used the A100 SXM4 version to perform the evaluation, and our team used the A100 PCIE version. The SXM4 version has a significantly higher rated power consumption (400W v.s. 250W for the PCIE version). Also, SXM4 has a different memory bandwidth. We re-examined the evaluation log and confirmed that in our evaluation environment described in L810-L812, the Spec-Bench toolkit works as expected to report faithful results in Table 1.
>
> 3. We re-ran the Medusa method on RTX-4090 GPU and reported the results in the revised submission (L330). **ParallelSpec still brought significant relative improvement (2.31$\times$ v.s. 1.63$\times$)** in terms of speedup ratio when evaluating with a different GPU.
>
> 4. For EAGLE ~10% difference with Spec-Bench Paper, **we think this is mainly due to the different hardware configurations we have.** We appreciate your understanding that, due to budget and practical constraints, we do not have access to the identical computational resources as the Spec-Bench team. This challenge is a common issue in the SPD community when reporting speedup results, as the essence of speculative decoding is to exploit redundant computation intensity in hardware accelerators.

---

> ### Author Response · Authors · 2024-11-22
> **Discussion (Part 2)**
>
> (To continue)
>
> 4. However, **since all the improvements in this paper were obtained using the same hardware environment, our comparisons are relatively fair.** We also notice that in the Spec-Bench latest benchmark, Table 5 and Table 6 also reported entirely different sets of speedup ratios when the only difference is the GPU used for benchmarking, indicating the hardware specifications are one of the major factors that impact reported speedup. We add explanations in L869 to discuss this issue.
>
> > Other action items (Q1-Q8)
>
> We are grateful to the comprehensive comments for our paper. We made the following changes to respond to your suggestions.
>
> 1. Figure 1 has been revised to highlight the difference between the two drafting styles.
> 2. We revise the claims in the abstract and at the end of the introduction to reflect relative improvements for both models.
> 3. Related Works and Background Sections have been updated to incorporate the suggested references and fix the notation issue.
> 4. We revise L395-L396 to reflect the details of SpS implementation. In short, we followed Spec-Bench to use Huggingface Transformers assisted_generation module to benchmark the SpS method, which has a heuristic-driven $\gamma$ adjusting approach implemented in https://github.com/huggingface/transformers/blob/4e90b99ed916300b80bac9db793f2a96b2a87122/src/transformers/generation/candidate_generator.py#L238-L262.
> ----
> Thanks again for your review to make this paper better. We hope our responses above could address your concerns and please free feel to leave more questions if you have. We are always happy to engage in discussions.

---

> > ### Comment · Reviewer_bvJa · 2024-11-24
> >
> > Thank you for your detailed response. I find it a bit confusing that your current revision has a mix of things that are actually in the paper and things that are just notes for the reviewers (E.g., figure 1's caption has the text "The Left part of the figure has been revised to highlight the difference between the two drafting styles.").
> >
> > In any case, thank you for explaining about the discrepancy between your reported results and the one from the SpecBench paper, and for explaining about the static token trees. I've decided to raise my score based on these clarifications and modifications to the paper.
> >
> > I still think the novelty is not high as it strongly relies on techniques that have been employed in the past (in Monea et al and Xia et al), and the improvements are not huge, but I appreciate that you’ve managed to incorporate these techniques into EAGLE and Medusa to improve speed ups.

---

> > > ### Author Response · Authors · 2024-11-24
> > >
> > > Thank you for increasing the overall rating! We hope our method could provide the community with valuable insights into incorporating parallel decoding into state-of-the-art draft-then-verify frameworks, and we hope our explanation regarding the novelty issue will help alleviate your concerns.

---

### Official Review · Reviewer_YbZP · 2024-11-02

**Soundness:** 2
**Presentation:** 4
**Contribution:** 2
**Rating:** 5
**Confidence:** 5

**Summary:**

This paper proposes ParallelSpec, which replaces the traditional autoregressive draft model in speculative decoding with parallel draft model to achieve better inference efficiency. Previous work such as Medusa already considers parallel draft model. The difference between ParallelSpec and Medusa is: Medusa trains additional heads to predict tokens at different positions, while ParallelSpec uses different mask tokens to do that. The authors claim their approach can maximally exploit the parameter sharing and hance is more effective. The experiment results in temperature=0 cases support this claim. But the experiments do not include the comparison with Medusa in temperature=1 cases. The main challenge in ParallelSpec is that it requires a different training mask and attention mechanism. The authors address this challenge reasonably.

**Strengths:**

1. The intuition of why ParallelSpec can accelerate EAGLE is convincing. Based on the experiment results shown in Table 1, although ParallelSpec generates less tokens per iteration, it improves the drafting efficiency via parallel drafing, so it improves the overall efficiency.

2. The paper is well-written and easy to understand.

3. The paper includes comprehensive discussion on how to integrate the proposed method with state-of-the-art framework such as EAGLE and Medusa.

**Weaknesses:**

1. One of my main concern is the comparison between the proposed method and Medusa. Although authors provides an intuitive explanation that ParallelSpec has better parameter sharing, I am not fully convinced. So I expect to see a comprehensive comparison between Medusa and ParallelSpec in the experiments. However, in Table 1, the comparison between Medusa and Medusa+ParallelSpec only covers the temperature=0, but not temperature=1.

2. In Table 1, I believe the two settings (temp=0 and temp=1) are equally important. But temp=1 only covers a subset of baselines and target models. So the evaluation is not comprehensive to me.

3. Although the proposed method could accelerate EAGLE, the improvement is somewhat marginal (only about 1.1$\times$ speed up). This limits the contribution of this paper.

**Questions:**

1. Can authors complete the evaluations in Table 1?

2. Is the code open-sourced?

3. The EAGLE uses autoregressive decoding as drafers. Is it possible to combine Medusa and EAGLE, meaning it also trains different decoding heads for different token positions?

4. It would be helpful if authors could provide detailed breakdown (e.g., draft time, verification time) of different methods. And compare how much does the propsoed method accelerate the draft time of EAGLE.

---

> ### Author Response · Authors · 2024-11-22
>
> Dear Reviewer YbZP,
>
> We appreciate your positive feedback on our work, and we would like to address your concerns regarding incomplete evaluations, reproducibility, the EALGE-Medusa combination, and a detailed latency breakdown of ParallelSpec.
>
> > (W1 & W2 & Q1) Incomplete evaluations on Medusa with temperature=1.0
>
> Good catch! When evaluating ParallelSpec performance, we generally follow the EAGLE-2 [1] evaluation protocol on what combinations of speculative decoding (SPD) frameworks and target models we need to choose. The Table 1 caption of EAGLE-2 paper mentioned that SPD methods like Medusa are excluded when temperture=1 because they “relax acceptance conditions under non-greedy settings, which do not guarantee lossless acceleration.” As such, we follow their practice and exclude these methods from Table 1 in our paper. Of course, we would happily provide additional results in the revision Table 1 when temperature=1.0. It is worth noting that since the lossless generation is no longer guaranteed, speedup ratios will be reported by comparing the generation speed (tokens/s) instead of end-to-end latency for these methods.
> Other missing baselines in the table are mainly due to the absence of corresponding target models, e.g., Medusa does not provide official checkpoints for Llama-2.
>
> > (Q2) Code availability
>
> Reproducibility is always our focus. The code will be publicly available upon the completion of the review process.
>
> > (Q3) Possibility to combine Medusa and EAGLE
>
> We believe that combining Medusa with an EAGLE-style approach—i.e., using multiple independent transformer decoder layers to predict different token positions—is certainly feasible. However, this approach runs counter to the design philosophy of ParallelSpec. Firstly, the primary motivation behind designing ParallelSpec was to avoid the memory overhead caused by Medusa-style multi-language model heads and to better leverage soft [MASK] tokens to guide the drafter to predict future tokens at different token positions, thereby achieving parameter sharing. Combining Medusa with EAGLE would undermine both of these assumptions, which is why this design was not considered within ParallelSpec.
> However, we acknowledge that this design could still be a valid alternative, provided the tradeoff between draft accuracy and memory overhead is carefully evaluated.
>
> > (Q4) Detailed breakdown of ParallelSpec latency
>
> We would like to first refer you to Appendix 1 for case studies on a comparison between Medusa and Medusa-ParallelSpec, where average draft latency and verification latency are denoted in Figure 5.
> For other methods, we report the per-token draft latency of Medusa, EAGLE, and ParallelSpec on vicuna-7b-v1.3, assuming the verification latency for these methods remains the same. Note that the draft latency for Medusa-ParallelSpec and EAGLE-ParallelSpec are identical since they share the same drafter architecture.
>
> | Method                          | Medusa | EAGLE | ParallelSpec |
> | ------------------------------- | ------ | ----- | ------------ |
> | Draft Latency Per Drafted Token | 1.5ms  | 2.3ms | 0.7ms        |
>
> Thanks again for your review in helping improve our paper, and please feel free to leave us more comments if you have; we are happy to engage in discussions.
>
> [1]: EAGLE-2: Faster Inference of Language Models with Dynamic Draft Trees (Li et al., EMNLP 2024)

---

> > ### Comment · Reviewer_YbZP · 2024-11-24
> >
> > I appreciate the authors’ feedbacks. I decide to maintain my current score of 6.

---

> > > ### Author Response · Authors · 2024-11-24
> > >
> > > Thanks for keeping the positive recognition for our work! Your comments are crucial in the discussion stage and contribute so much to the final revision.

---

### Official Review · Reviewer_yjYC · 2024-11-03

**Soundness:** 3
**Presentation:** 3
**Contribution:** 3
**Rating:** 6
**Confidence:** 3

**Summary:**

The paper introduces a parallel drafting model as an alternative to traditional auto-regressive speculative decoding (SD) drafters for large language models (LLMs). Traditional SD models often draft tokens sequentially, resulting in latency that scales linearly with sequence length. PARALLELSPEC innovates by using a parallel drafter that predicts multiple tokens in a single forward pass, achieving notable reductions in draft latency (up to 62%) and overall speedups of up to 2.84x on the LLaMA-2-13B model. The model’s integration into frameworks like Medusa and EAGLE showcases its compatibility and effectiveness, while a group-wise training strategy ensures alignment with the target model’s output distribution. This method maintains theoretical losslessness through a rejection sampling algorithm, preserving the original LLM distribution.

**Strengths:**

+ The paper’s parallel drafter introduces a novel alternative to sequential SD models, advancing speculative decoding with efficient multi-token generation.
+ Empirical evaluations across multiple benchmarks and models, along with sound theoretical justifications, confirm the validity of the approach.
+ The work has strong implications for real-time, large-scale LLM applications, especially with the demonstrated integration into established frameworks like Medusa and EAGLE.
+ By employing rejection sampling, PARALLELSPEC maintains a lossless decoding process, preserving the original output distribution of the target model.

**Weaknesses:**

- Unlike speculative decoding approaches that can use pre-trained models or lightweight modifications, PARALLELSPEC requires a dedicated training process for the parallel drafter. This additional training step may increase setup time and computational cost, which could limit the method’s immediate applicability in certain use cases.
- The absence of publicly available code for the training and experimental setups raises concerns about reproducibility. Without the code, it may be challenging for others in the community to replicate the experiments or validate the results independently, limiting the impact and accessibility of this work.
- The group-wise parallel training could use clearer explanations or diagrams, particularly the role of attention masks and mask tokens in aligning the drafter with the target model’s distribution.
- Handling of Token Uncertainty: While effective in low-temperature settings, PARALLELSPEC may experience reduced alignment at higher temperatures. Further discussion on managing token misalignment under these conditions would enhance robustness, particularly compared to EAGLE's feature-based uncertainty handling.

**Questions:**

1. Could the authors elaborate on the sensitivity of PARALLELSPEC’s performance to the number of tokens generated in parallel (i.e., the parameter k)? Are there guidelines for selecting k based on specific model or task characteristics?
2. How does PARALLELSPEC’s memory usage compare with traditional auto-regressive drafters, particularly in large models? Does the use of multiple [MASK] tokens introduce significant memory overhead?
3. In high-temperature settings, where acceptance rates may decrease, what are possible strategies to maintain drafter alignment without significantly impacting speedup?

---

> ### Author Response · Authors · 2024-11-22
>
> Dear Reviewer yjYC,
>
> Thank you for your insightful comments and your appreciation of the novel design in ParallelSpec and its strong implications for real-world LLM applications. We would like to address your concerns below.
>
> > ParallelSpec requires a dedicated training process that limits its immediate applicability.
>
> As a speculative decoding framework that requires an additional training stage to align drafter distribution to its target model, we agree that such an alignment stage prevents the immediate application of ParallelSpec. However, training-free methods such as conventional speculative sampling (SpS) usually have lower speedup ratios compared to those methods that need extra alignment, as indicated in Table 1, where Hydra, Medusa, EAGLE, and ParallelSpec obtain preferable performance. Therefore, we think **a dedicated training process could be good for applications that prefer higher speedup to immediate applicability if non-trivial performance improvement can be observed.**
>
> > Reproducibility Issue
>
> Thanks for raising the reproducibility issue. We describe the experimental setting and hyper-parameters in Section 5.1 and Appendix A.2. The code will be publicly available upon the completion of the review process.
>
> > Better explanation of group-wise parallel training
>
> We apologize for not clearly describing the group-wise parallel training strategy. The motivation for such a design is to ensure the model behaves the same in both the alignment training and drafter inference stages. The attention mask in Figure 3 significantly accelerates the training by packing multiple parallel groups into ONE single training step and uses a customized attention mask to prevent information leakage across different groups. The special input token organization and position indices in the right part of Figure 2 ensure identical behaviors between training and inference.
>
> > $K$ sensitivity and group-size $K$ selection guideline
>
> We kindly refer you to Figure 4, which illustrates $K$ sensitivity in data from four domains. Following the common practice of how auto-regressive speculative sampling methods determine draft length $\gamma$, we choose the group size $K$ by observing the end-to-end speedup ratio on validation sets in the target domains.
>
> > Memory overhead of [MASK] tokens
>
> Good question! We use the Deepspeed Memory Profiler to measure the max memory allocated for Medusa, EAGLE, and EAGLE-Parallel with the same target model vicuna-7b-v1.3 during the benchmark on Spec-Bench, with fp16 precision and batch_size=1.
>
> | Naive Decoding | Medusa             | EAGLE              | EAGLE-ParallelSpec |
> | -------------- | ------------------ | ------------------ | ------------------ |
> | 13,716MB (0MB) | 17,184MB (+3468MB) | 16,486MB (+2770MB) | 16,493MB (+2777MB) |
>
> When benchmarking with the same sequence, the only memory overhead brought by ParallelSpec is the $K$ [MASK] token embeddings and associated intermediate activations, which are negligible compared to the target model and drafter model size. Thanks to the transformer-based drafter model design, ParallelSpec also demonstrates superior memory consumption compared to the Medusa multi-head drafter design.
>
> > High-temperature settings speedup impact
>
> While ParallelSpec does not have explicit designs in high-temperature setups, we are glad to discuss some strategies to maintain drafter alignment without significantly impacting speedup.
> For example, in the knowledge distillation setting, we can consider using moving earth distance to better align the draft with the target model under high temperatures. Specifically, given the p_draft, q_target, and T(randomly sampled from [0.8, 1]), where p and q are logit distributions of draft and target models, we can modify the distillation loss so that the expectation of acceptance rate E_{T,t} where t is the token could be maximized. In this way, the alignment of the two models could be maximized during the distillation process via sampling from higher temperatures from [0.8,1].
>
> We sincerely appreciate your review in helping improve our paper and hope our response will be satisfactory. Please feel free to leave us more comments, and we will be happy to engage in discussions.

---

> ### Comment · Reviewer_yjYC · 2024-11-25
>
> Thanks for the detailed response. I'm increasing these scores:
> - Soundness: 3: good
> - Contribution: 3: good
> - Rating: 6: marginally above the acceptance threshold

---

> > ### Author Response · Authors · 2024-11-25
> >
> > We sincerely appreciate your positive feedback on our work! Your perspectives on memory overhead and high-temperature settings have helped improve the paper. If feasible, we would be grateful if you could update your review to reflect your recognition of our contributions.

---

### Comment · Area_Chair_x3WK · 2024-11-24
**Reminder: Author-Reviewer Discussion Period Ends Soon**

Dear Reviewers

This is a reminder that the author-reviewer discussion period will end on Nov 26 AoE.

Your engagement during this phase is critical for providing valuable feedback and clarifications. If you have any remaining questions or comments, please take a moment to participate before the deadline.

Thank you for your contributions to this important process.

AC

---

### Meta-Review · Area_Chair_x3WK · 2024-12-19

**Metareview:**

(a) Summary of Scientific Claims and Findings

This paper introduces ParallelSpec, which leverages mask tokens to enable the drafter model to predict multiple tokens simultaneously within the speculative decoding framework.

(b) Strengths of the Paper

1. Demonstrates notable speed improvements for speculative decoding frameworks.

2. Provides robust empirical validation supported by strong theoretical foundations.

(c) Weaknesses and Missing Elements

1. The approach is considered incremental, bearing significant resemblance to prior works (e.g., Xia et al., Stern et al.).

2. Lacks evaluations on larger models, which are crucial for assessing the effectiveness of speculative decoding techniques.

(d) Decision and Rationale
Reviewers expressed significant concerns regarding the novelty of this work, citing substantial similarities with existing approaches.

**Additional Comments On Reviewer Discussion:**

Despite the authors addressing numerous concerns raised by the reviewers during the discussion phase, the issue of novelty remains unresolved and is a significant concern.

---

### Decision · Program_Chairs · 2025-01-22

Reject